# Towards practical dynamic induction control of wind farms: analysis of optimally controlled wind-farm boundary layers and sinusoidal induction control of first-row turbines

Wim Munters and Johan Meyers

Department of Mechanical Engineering, KU Leuven, Celestijnenlaan 300A, 3001 Leuven, Belgium

*Correspondence to:* wim.munters@kuleuven.be

**Abstract.** Wake interactions between wind turbines in wind farms lead to reduced energy extraction in downstream rows. In recent work, optimization and large-eddy simulation were combined with optimal dynamic induction control of wind farms to study the mitigation of these effects, showing potential power gains of up to 20% (Munters & Meyers 2017 *Phil Trans R Soc A* **375**, 20160100, doi:10.1098/rsta.2016.0100). However, the computational cost associated with these optimal control simulations impedes practical implementation of this approach. Furthermore, the resulting control signals optimally react to the specific instantaneous turbulent flow realizations in the simulations, so that they cannot be simply used in general. The current work focuses on the detailed analysis of the optimization results of Munters & Meyers, with the aim to identify simplified control strategies that mimic the optimal control results and can be used in practice. The analysis shows that wind-farm controls are optimized in a parabolic manner with little upstream propagation of information. Moreover, turbines can be classified into first-row, intermediate-row, and last-row turbines based on their optimal control dynamics. At the moment, the control mechanisms for intermediate-row turbines remain unclear, but for first-row turbines we find that the optimal controls increase wake mixing by periodic shedding of vortex rings. This behavior can be mimicked with a simple sinusoidal thrust control strategy for first-row turbines, resulting in robust power gains for turbines in the entrance region of the farm.

## 1 Introduction

Wake interactions between turbines within a wind farm cause reduced power extraction and increased turbine loading in downstream rows. The current control paradigm in such farms optimizes performance at the wind-turbine level and does not account for these interactions, resulting in sub-optimal wind-farm efficiency. In contrast, control strategies at the farm level allow to influence wake interaction and promise to improve overall wind-farm performance by improving wind conditions for downstream turbines. This can be achieved by redirecting propagating wakes (yaw control, see e.g. Fleming et al., 2014; Gebraad et al., 2016; Campagnolo et al., 2016) or by affecting the induced wake velocity deficits (axial induction control, see e.g. Nilsson et al., 2015; Annoni et al., 2016; Bartl and Sætran, 2016). A more exhaustive survey of wind-farm control in a broader context can be found in Knudsen et al. (2015) and Boersma et al. (2017).

In contrast to the studies cited above, that all focus on static setpoint optimization of wind farms, Goit and Meyers (2015) introduced a dynamic induction control approach based on large-eddy simulations (LES) and adjoint gradient optimization.

In this study, individual turbines were used as dynamic flow actuators that influence the wind-farm boundary layer flow in such a way as to optimize aggregate wind-farm power extraction. The methodology was applied to the asymptotic case of a fully-developed 'infinite' aligned wind farm, and power gains of about 16% were quantified. Later, this approach was also used in induction control studies of wind farms with entrance effects in Goit et al. (2016) and, more recently, in Munters and Meyers (2017), where similar gains in the order of 15%–20% were achieved. It is important to note that the computational cost of this LES-based dynamic induction control methodology is orders of magnitude too high for direct implementation as a practical control strategy. However, the methodology allows to assess the theoretical potential for wind-farm control, and increased understanding of the flow physics can lead to simplified control strategies that can be applied in practice.

Recently, the methodology of Goit and Meyers (2015) was generalized to include dynamic yaw control in Munters and Meyers (2018). In this study, induction control and yaw control were compared for a relatively small aligned wind farm, and yaw control was found to yield higher power gains for this setup. Furthermore, the high potential of combined induction and yaw control was quantified, and analysis of the yaw control signals allowed to identify practical simplified dynamic yaw control strategies. The search for similar practical control strategies for induction control has remained unsuccessful to date.

The current paper presents efforts on understanding optimal control dynamics observed in the optimal induction control simulations by Munters and Meyers (2017) (further denoted as MM17). The outline of the paper is as follows: first, Sect. 2 discusses the numerical setup and optimal control simulations of MM17 that will be further analysed in the current paper. Section 3 presents an analysis of the control and thrust force dynamics, and performs some numerical experiments to elucidate characteristics of the optimal controls. It will be shown that the coherent behavior of turbines situated in the first row of the wind farm stands out from their downstream counterparts. Hereafter, Sect. 4 identifies the shedding of vortex rings from the first row based on a flow visualization. Further, a simple sinusoidal thrust control approach is presented that successfully mimics this process with a robust increase in power extraction extraction in the second row. Next, Sect. 5 shortly discusses the behavior of the intermediate rows, i.e. turbines which have both upstream and downstream neighbors, for which similar simple control strategies remain elusive thus far. In conclusion, Sect. 6 summarizes the main findings of this paper.

## 2    Description of optimal control simulations in MM17

The current section describes the optimal control simulations performed by MM17, the results of which are further analyzed in the current paper. First, the methodology is introduced. Then, the numerical setup is detailed. Afterwards, the optimization results on power extraction and time-averaged flow field features are discussed.

### 2.1    Control methodology

A schematic overview of the wind-farm control methodology is shown in Fig. 1. Figure 1a illustrates the control block diagram: an iterative optimization loop updates the wind-farm control vector $\varphi(t)$ until a set of optimized controls $\varphi^{\bullet}(t)$ is found. This optimization is based upon an unsteady turbulence-resolving LES wind-farm flow model, and sensitivities of the cost functional $\mathscr{J}$ (i.e. the total wind-farm power extraction) are calculated using an adjoint formulation of this flow model. In this way, a

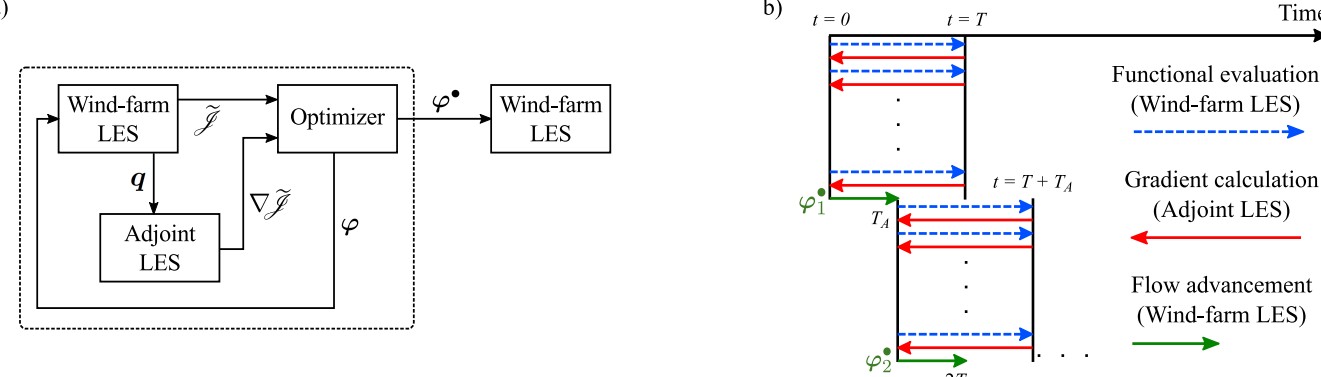

**Figure 1.** Schematic overview of wind-farm optimal control methodology from MM17. *a)* Control block diagram with adjoint gradient-based optimization and LES flow models illustrating data flow of (optimal) controls $\boldsymbol{\varphi}^{(\bullet)}$, system state $\boldsymbol{q}$, cost functional $\mathscr{J}$ and its gradient $\nabla\mathscr{J}$. *b)* Receding horizon framework subdividing time into discrete flow advancement windows of length $T_A$ with prediction horizon $T$. Each arrow represents a forward or adjoint LES. Every window consists of an optimization stage (blue and red lines) follow by a flow advancement stage with optimal controls $\boldsymbol{\varphi}^{\bullet}$ (green lines).

priori simplifications to the turbulent boundary layer and wake representation are avoided as much as possible, and the control signals are designed in a such a way that turbines actively tap into the dynamics of the turbulent flow. The optimization is performed using a receding-horizon control framework, as illustrated in Fig. 1b. In this framework, wind-farm controls $\boldsymbol{\varphi}(t)$ are optimized for a finite time horizon $T$, involving a sequential set of LES and adjoint LES simulations. Upon convergence of

5   the optimization, optimized control signals are applied in a flow advancement simulation for a time $T_A < T$, after which a new optimization window is initiated.

Within each optimization window, the total wind-farm power is optimized by solving the following partial-differential-equation-constrained optimization problem:

$$\underset{\boldsymbol{\varphi},\boldsymbol{q}}{\text{minimize}} \qquad \mathscr{J}(\boldsymbol{\varphi},\boldsymbol{q}) = -\int_0^T \sum_{i=1}^{N_t} P_i \, \mathrm{d}t \tag{1}$$

10   s.t.

$$\frac{\partial \widetilde{\boldsymbol{u}}}{\partial t} + \left(\widetilde{\boldsymbol{u}} \cdot \nabla\right)\widetilde{\boldsymbol{u}} = -\nabla(\widetilde{p}+\widetilde{p}_\infty)/\rho - \nabla \cdot \boldsymbol{\tau}_{sgs} + \sum_{i=1}^{N_t} \boldsymbol{f}_i \qquad \text{in } \Omega \times (0,T], \tag{2}$$

$$\nabla \cdot \widetilde{\boldsymbol{u}} = 0 \qquad\qquad\qquad \text{in } \Omega \times (0,T], \tag{3}$$

$$\tau \frac{\mathrm{d}\widehat{C}'_{T,i}}{\mathrm{d}t} = C'_{T,i} - \widehat{C}'_{T,i} \qquad\qquad i = 1\ldots N_t \text{ in } (0,T], \tag{4}$$

$$0 \leq C'_{T,i} \leq C'_{T,\text{max}} \qquad\qquad i = 1\ldots N_t \text{ in } (0,T], \tag{5}$$

The cost functional that is optimized in (1) is the total wind-farm energy extraction over time horizon $T$. The control variables

15   are the time-dependent thrust coefficient setpoints $C'_{T,i}$ of every turbine $i$ $(=1\ldots N_t)$, i.e. $\boldsymbol{\varphi} = [C'_{T,1}(t),\ldots,C'_{T,N_t}(t)]$, and the

state variables are denoted as $\boldsymbol{q} = [\widetilde{\boldsymbol{u}}(\boldsymbol{x},t); \widetilde{p}(\boldsymbol{x},t); \widehat{C}'_{T,1}(t),\ldots,\widehat{C}'_{T,N_t}(t)]$, with $\widetilde{\boldsymbol{u}}$ the filtered velocity, $\widetilde{p}$ the filtered pressure, and $\widehat{C}'_{T,i}$ the (time-filtered) thrust coefficient for turbine $i$ (see below).

The filtered Navier–Stokes momentum and continuity state equations in (2) - (3) are solved using an in-house LES solver (see, e.g. Calaf et al., 2010; Meyers and Meneveau, 2010; Goit et al., 2016 for a detailed discussion of the solver). The time-filtering state equation in (4) applies a one-sided exponential time filter to the thrust coefficient setpoints $C'_{T,i}$ with a characteristic wind-turbine response timescale $\tau$ (Munters and Meyers, 2016). Finally, the box constraints in (5) limit the variations in the turbine thrust coefficients to technically feasible values.

The forces exerted by turbine $i$ on the boundary-layer flow are parametrized using a standard non-rotating actuator disk model as $\boldsymbol{f}_i(\boldsymbol{x},t) = -(1/2)\widehat{C}'_{T,i}(t)V_i(t)^2\mathscr{R}_i(\boldsymbol{x})\boldsymbol{e}_{\perp,i}$, where $\mathscr{R}_i$ is a smoothed representation of the geometric footprint of the turbine on the LES grid and $\boldsymbol{e}_{\perp,i}$ is the rotor-perpendicular vector. Further, the disk-averaged velocity is defined as $V_i = (1/A_i)\int_\Omega \mathscr{R}_i(\boldsymbol{x})\widetilde{\boldsymbol{u}}\cdot\boldsymbol{e}_{\perp,i}\,\mathrm{d}\boldsymbol{x}$, with $A_i$ the rotor disk area. Mechanical power captured by the wind turbine is calculated as $P_i = (1/2)C'_{P,i}(t)V_i(t)^3 A_i$, with $C'_{P,i} = 0.9C'_{T,i}$, resulting from a fit of LES results to momentum theory, eliminating the overprediction of wind-turbine power on typical wind-farm LES grid resolutions (Munters and Meyers, 2017).

## 2.2 Case setup

The wind farm considered in MM17 has an aligned pattern of 12 rows by 6 columns. The wind turbines have a hub height $z_h = 100$ m with a rotor diameter $D = 100$ m, and are spaced apart by $6D$ in both axial and transversal directions. The flow is simulated on a domain of $10 \times 3.6 \times 1$ km$^3$, discretized on a simulation grid of $384 \times 192 \times 144$ grid points. A snapshot of the streamwise velocity field is shown in Fig. 2. The wind farm was controlled over a total of $N_A = 15$ time windows with a prediction horizon $T = 240$ s (i.e. the time it takes for the flow to pass four rows of turbines) and a flow advancement time of $T_A = T/2 = 120$ s, resulting in a total control time $T_{\mathrm{tot}} = N_A T_A = 30$ minutes.

A conventionally (greedily) controlled wind farm with steady $C'_T = 2$ was defined as a reference case. Note that this would correspond to a farm with ideal turbines for which generator torque is being controlled dynamically to track the maximum power point at the Betz limit perfectly. In a real turbine controller this may, e.g., be achieved with the extremum seeking control proposed by Ciri et al. (2017). Several different optimal control cases were defined, based on the wind-turbine response time $\tau = 0$, 5, or 30 s (instantaneous, fast, or slow response, see Eq. 4) and the maximal thrust coefficient $C'_{T,\mathrm{max}} = 2$ or 3 , with thrust forces that can respectively only be reduced (underinductive), or also increased (overinductive) compared to the Betz optimum at $C'_T = 2$ (see Eq. 5). Cases are denoted as C<X>t<Y>, where X and Y represent $C'_{T,\mathrm{max}}$ and $\tau$ respectively, e.g. C3t30 for the case with $C'_{T,\mathrm{max}} = 3$ and $\tau = 30$ s. The choice of (and sensitivity to) setup parameters is further elaborated in MM17.

## 2.3 Simulation results

Figure 3 illustrates the energy extraction of the optimally controlled wind farm cases, normalized by the greedy reference control case. Figure 3a shows that the adjoint LES-based control approaches achieves energy gains ranging from a minor 2% in the most restrictive C2t30 case to over 20% in the C3t0 case. From Fig. 3b it can be seen that, for all cases except C3t30,

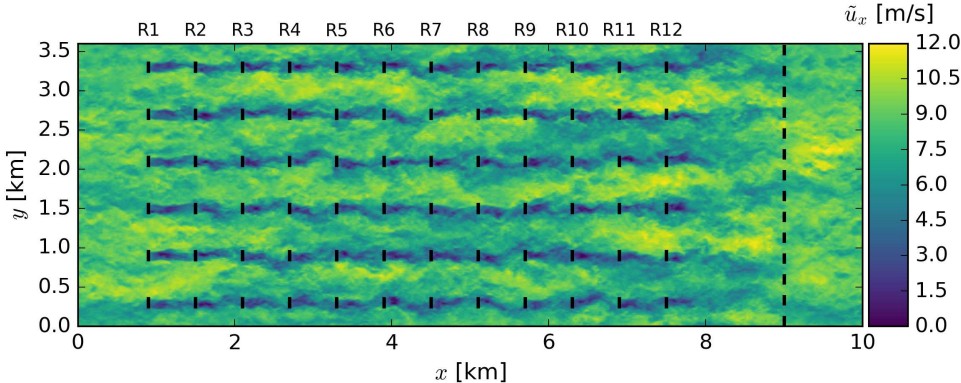

**Figure 2.** Instantaneous streamwise velocity $\widetilde{u}_x$ for the $12 \times 6$ aligned wind farm considered in MM17. Black lines indicate wind-turbine locations. The black dashed line near the end of the domain indicates a buffer region used for imposition of inflow boundary conditions. Figure originally published in Munters and Meyers (2017) under a CC-BY 4.0 license.

power is curtailed in the first row to a limited degree, whereas the downstream rows compensate for this loss by extracting significantly more energy. Furthermore, not taking into account the first row, the last row achieves the highest energy extraction in every case, as it can act greedily without compromising power extraction in downstream neighbors.

In the remainder of this section, time-averaged wind-farm flow properties will be investigated. Here and throughout the remainder of this paper, we focus on case C3t5, as it produces similar energy gains as the highest-yield case C3t0 (see Fig. 3a), but achieves this with smoother thrust coefficient signals. In the following discussion, the time averaging operation is denoted by an overline, and flow field variables are decomposed into mean and fluctuating components as $\widetilde{u} = \overline{\widetilde{u}} + \widetilde{u}' \equiv \widetilde{U} + \widetilde{u}'$. Figure 4 illustrates time-averaged flow field quantities of the reference case (left panels, a1 – g1) and the differences between the optimized C3t5 case and the reference case (right panels, a2 – g2). Simulation results are averaged over the different columns and are shown as either topviews at hub height (Figs. 4b,f,g) or sideviews through a turbine column (Figs. 4a,c,d,e).

Figures 4a and 4b illustrate sideviews and topviews of the axial velocity throughout the wind farm. It can be seen that downstream turbines in the controlled case experience consistently higher incoming velocities, which explains the increased energy extraction discussed above. Furthermore, a larger drop in streamwise velocity over the turbine disk can be observed, most notably in the first-row turbines, indicating deeper wakes with enhanced recovery before hitting the next row of turbines. Furthermore, it can be observed that the axial velocity in the flow above and besides the wind-turbine column is reduced, indicating that the mean-flow kinetic energy is depleted in these regions, to the benefit of the flow passing through the wind turbines.

Figure 4c shows sideviews of turbulence kinetic energy $k$. The figure shows an increase in turbulence throughout the entire wind farm, spreading to the internal boundary layer above the turbines. Note specifically the sharp increase in turbulence in the core wake region behind the first-row turbine, for which an enhanced recovery was found as discussed above. The turbulence

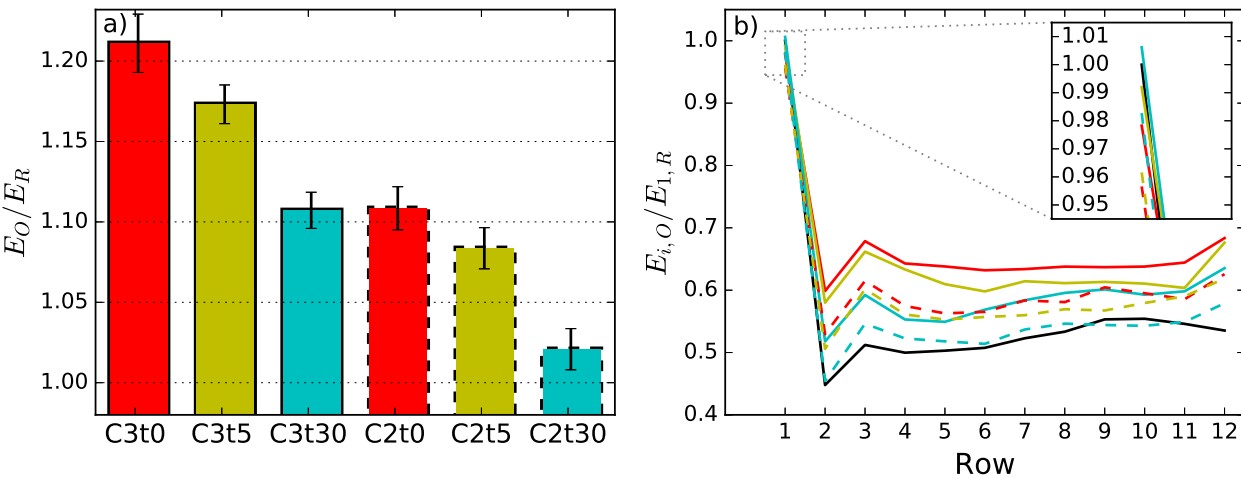

**Figure 3.** Energy extraction $E_O$ of optimally controlled wind-farm cases from MM17, normalized by a greedy reference case $E_R$. *a)* Total energy extraction. Error bars indicate confidence intervals of $\pm 2$ standard deviations. *b)* Energy extraction by row, normalized by first-row reference power. C3t0, red line; C3t5, yellow line; C3t30, blue line; C2t0, red dashed line; C2t5, yellow dashed line; C2t30, blue dashed line. Figure originally published in Munters and Meyers (2017) under a CC-BY 4.0 license.

intensity $TI \equiv (2k/3)^{1/2}/U_\infty$ at hub height (not shown in the figure) is 10% at the inlet for both the reference case and the controlled case. The combination of reduced near-wake mean velocities and increased velocity fluctuations in the controlled case increase local $TI$ in the turbine wakes (ranging from $\approx 2\%$-points in the wakes of middle rows to $\approx 12\%$-points in the first and last rows). This increase in turbulence intensity dissipates to below 1%-point difference at $10D$ downstream of the last

row.

Figure 4d and 4e show sideviews through the rotor centerline of top-down turbulence and mean-flow transport of axial momemtum, i.e. $-\widetilde{u'_x u'_z}$ and $-\widetilde{U}_x\widetilde{U}_z$ respectively. It can be seen that, although mean-flow vertical transport is virtually unaffected, turbulence top-down transport of axial momentum is increased significantly in the upper part of the wakes, indicating increased turbulent mixing with the internal boundary layer above the wind-farm canopy. The effect is somewhat more pronounced in

the wake behind the first row of turbines, for which also a slight increase in upwards transport of momentum can be observed in the lower part of the wake.

Figure 4f and 4g show planviews at hub height of transversal turbulence and mean-flow transport of axial momemtum, i.e. $\widetilde{u'_x u'_y}$ and $\widetilde{U}_x\widetilde{U}_y$ respectively. The sign convention is such that positive values correspond to transport in the positive $y$ direction in the figure. A slight increase in turbulent transversal transport towards the wake centerline can be observed behind every

row. The mean-flow transversal momentum transport into the wake region is increased significantly behind the first two turbine rows. Downstream of these rows, the difference between these cases and the reference case is far less coherent. The latter can

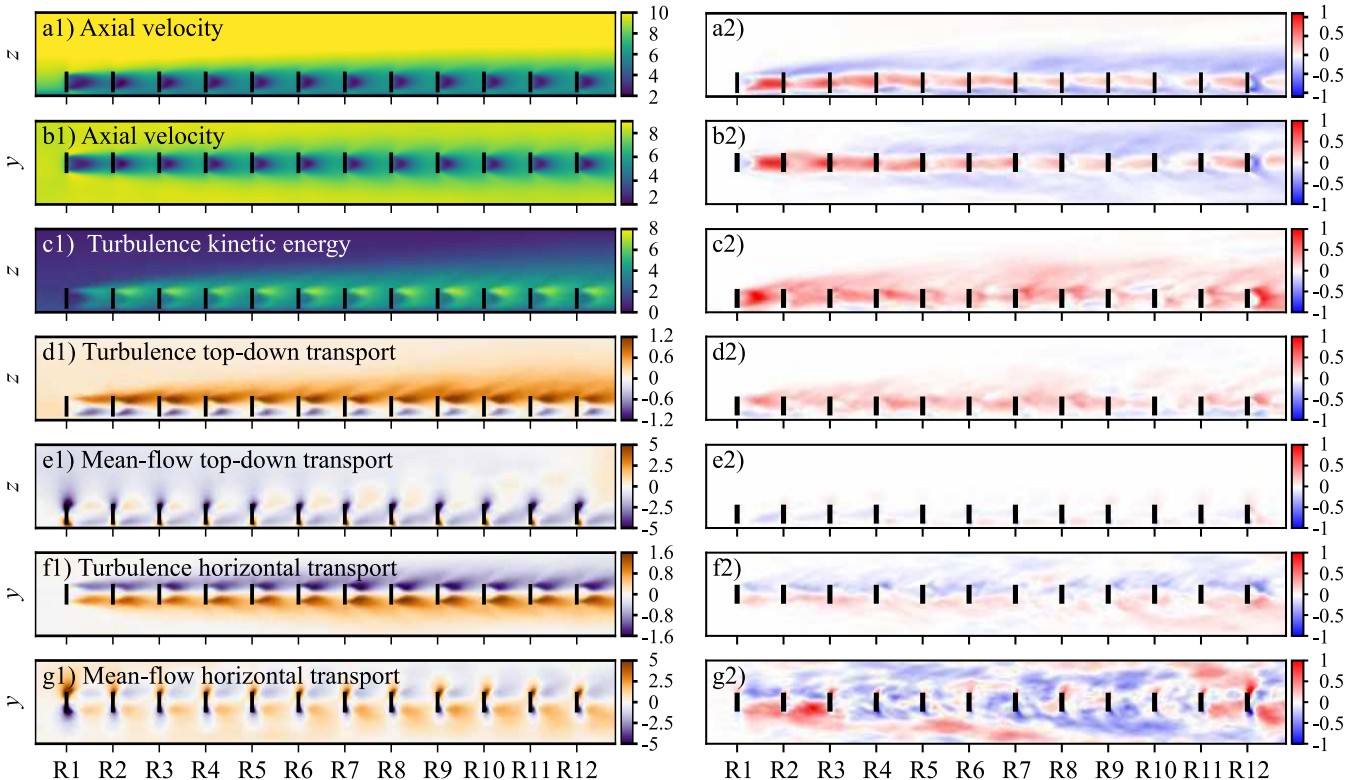

**Figure 4.** Time-averaged flow field quantities of simulation results from MM17. *Left (1):* Time averages of reference simulations with $C'_T = 2$ for all turbines. *Right (2):* Difference $\Delta$ between reference and optimal control (C3t5) simulation, defined as $\Delta = X_{\mathrm{C3t5}} - X_{\mathrm{REF}}$ for any variable $X$. *a):* Axial velocity $U_x$ (planview at hub height). *b):* Axial velocity $U_x$ (sideview through turbine column). *c):* Turbulence kinetic energy $k = (1/2)(\overline{\widetilde{u'_x}\widetilde{u'_x}} + \overline{\widetilde{u'_y}\widetilde{u'_y}} + \overline{\widetilde{u'_z}\widetilde{u'_z}})$ (sideview). *d):* Turbulence top-down transport $-\overline{\widetilde{u'_x}\widetilde{u'_z}}$. *e):* Mean-flow top-down transport $-\widetilde{U}_x\widetilde{U}_z$ (sideview). *f):* Turbulence horizontal transport $\overline{\widetilde{u'_x}\widetilde{u'_y}}$ (planview). *g):* Mean-flow horizontal transport $\widetilde{U}_x\widetilde{U}_y$ (planview). Black lines indicate wind turbine locations. Simulation results are averaged over the six different wind turbine columns.

be explained by the fact that $\widetilde{U}_x$ in the intercolumn channels starts to deviate significantly from the reference case as shown in Fig. 4b.

The analysis of flow features given above indicates that the optimal controls in case C3t5 influence the wind-farm flow field in such a way as to provide better flow conditions for downstream turbines. Increased axial velocities are observed for all downstream turbines, and enhanced momentum transport towards the turbine region is achieved. Furthermore, many of the observed flow features are most salient for the first row turbines. In the following section, the optimized thrust coefficients themselves will be investigated. It will be shown that, also from a controls perspective, first-row turbines stand out from their downstream counterparts.

## 3 Thrust coefficient analysis and numerical experiments

The current section focuses on the optimal thrust coefficients generated by the optimal control simulations in MM17, and performs numerical experiments to uncover some of the characteristics of these control signals. Note that the conclusions drawn within this section should be interpreted as observations of the current C3t5 optimal control cases, given specific wind-farm layout and flow conditions, and hence cannot just be generalized for any wind-farm control in general.

First, the thrust coefficient signals themselves are analyzed in Sect. 3.1. Second, the optimized thrust coefficients are applied only to subsets of turbine rows in Sect. 3.2. In this way, the interdependency of optimized thrust coefficients in different rows can be evaluated. Third, additional optimal control simulations, in which only one single active row is optimized, are discussed in Sect. 3.3. These optimizations provide an indication on how increased power potential is distributed among the rows, and allows to compare the resulting single-row optimized thrust coefficients with the fully cooperative coefficients from case MM17. Fourth, Sect. 3.4 evaluates the dependency of optimized thrust coefficients on the actual turbulent flow realization. Finally, Sect. 3.5 discusses the main conclusions from the abovementioned sections, and summarizes the lessons learned.

### 3.1 Analysis of thrust coefficient signals

Figure 5 illustrates the time evolution of some of the thrust coefficients $\widehat{C}'_T$ in the C3t5 case. The figure shows that, for all rows but the last one (i.e. R12), $\widehat{C}'_T$ varies significantly in time, and that the amplitudes and frequencies of these variations are somewhat higher in the upstream rows of the farm. In contrast, row 12 features only minor unsteadiness at lower frequencies, and has an increased mean value of $\widehat{C}'_T$. This relatively steady behavior of the last row can be explained by the fact that there are no further downstream turbines that can benefit from row 12 actively influencing local flow conditions, hence the row optimizes its own power only. The increase in mean $\widehat{C}'_T$ in row 12 can be explained based on Fig. 6, which shows the power extraction as a function of steady $\widehat{C}'_T$ for unwaked turbines, subject to identical turbulent inlet as in case C3t5. Although momentum theory predicts maximal power extraction for steady uniform inflow at $\widehat{C}'_T = 2$, the actual optimal steady value for the ADM at the current spatial resolution lies somewhat higher at $\widehat{C}'_T \approx 2.4$, for which power extraction is about 1.4% higher than at $\widehat{C}'_T = 2$. This behavior is related to the overprediction in ADM power due to the diffuse turbine representation on typical simulation grids: the mass flow through the rotor disk at $\widehat{C}'_T > 2$ is slightly too high compared to momentum theory, resulting in a shift of optimal $\widehat{C}'_T$ towards somewhat higher values. Although the linear fit $C'_P = a\widehat{C}'_T$ introduced in Munters and Meyers (2017), Appendix A eliminates the error in maximal power extraction, it does not correct the value of the optimal $\widehat{C}'_T$ (note that this could be achieved through a more complex relation between $C'_P$ and $\widehat{C}'_T$). Returning to the more complex thrust coefficients in the other rows it is worth noting that, based on the current dataset, no statistically significant correlations between thrust coefficients of different turbines could be found. Furthermore, attempts towards linking thrust coefficient dynamics to upstream flow measurements (e.g. velocities, shear or kinetic energy) through linear regression models and random forest regressors have been unsuccessful to date.

Figure 7a,b shows row-averaged power spectral densities of the thrust forces and thrust coefficients respectively. The figure shows that the variances of both the thrust coefficients and their resulting forces are highest in row 1. Further downstream,

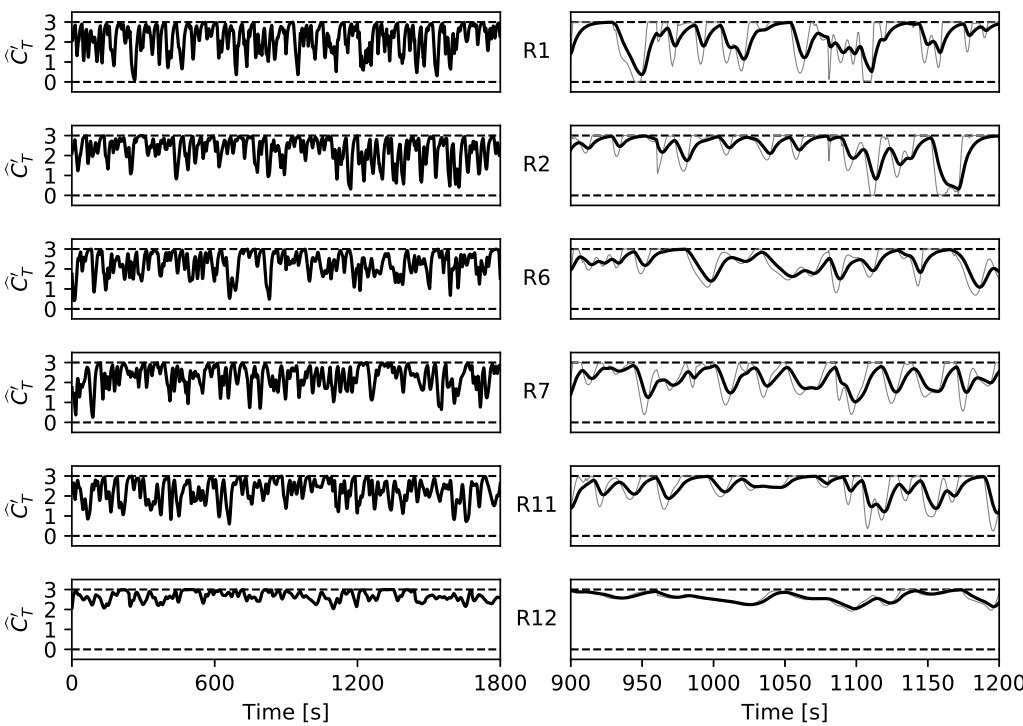

**Figure 5.** Time evolution of the thrust coefficient $\widehat{C}'_T$ for a selection of optimally controlled turbines of case C3t5. *Left:* Total time horizon. *Right:* Zoomed view, including setpoint $C'_T$ in gray.

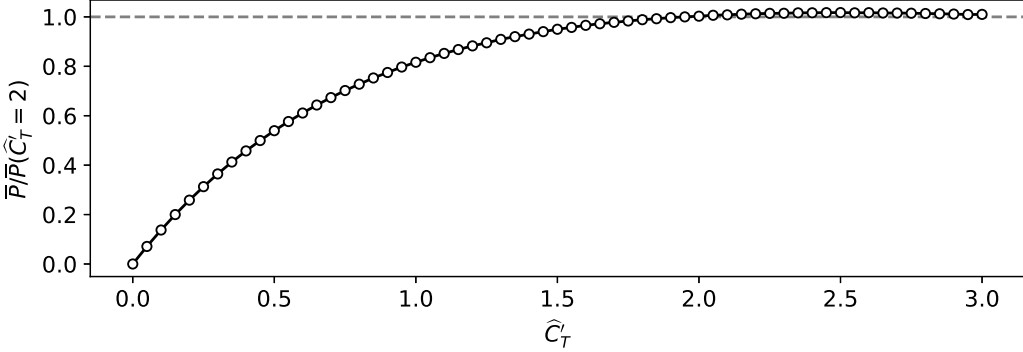

**Figure 6.** Normalized power extraction as a function of steady thrust coefficient $\widehat{C}'_T$ for wind turbines subject to the same freestream turbulent inflow as in case C3t5. Every dot corresponds to one LES.

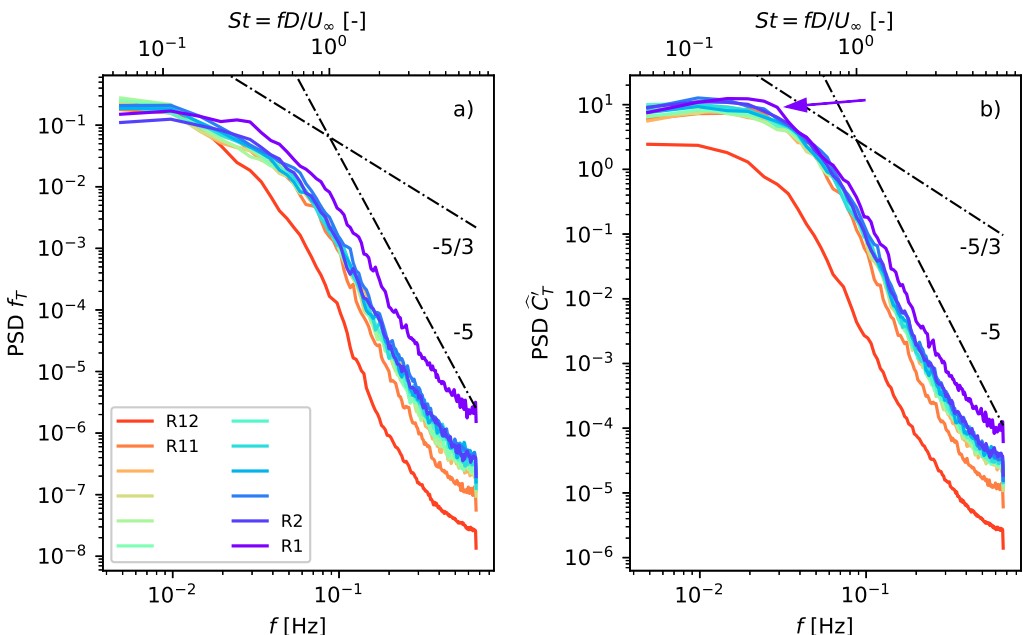

**Figure 7.** Power spectral density (PSD) estimates of the row-averaged thrust force $f_T$ (*a*) and thrust coefficient $\widehat{C}'_T$ (*b*) as a function of frequency $f$ (bottom axis) and non-dimensional Strouhal number $St$ (top axis).

rows 2 to 11 have very similar spectral behavior, and row 12 shows significantly lower variability. The high-frequency slopes of around $-5$ observed both for $f_T$ and $\widehat{C}'_T$ indicate that force variability on short timescales is caused mainly by thrust coefficient variations, whereas the slower thrust force dynamics tend more to a $-5/3$ slope, suggesting that these are governed by the unsteadiness in the turbulence instead. Note that, even though the spectra for all rows except row 12 collapse at frequencies below $0.05$ Hz, the first-row spectrum shows a small peak at $f \approx 0.02\ldots0.03$ Hz ($fD/U_\infty \approx 0.2\ldots0.3$) as indicated by the purple arrow. It will be shown later in this paper that variations in the thrust coefficient around this frequency are directly related to increased power extraction.

### 3.2 Application of optimal thrust coefficients to subsets of turbine rows

In order to further study how the optimal controls increase overall wind-farm power, Fig. 8 shows power extraction resulting from applying a subset of the optimal controls to specific turbine rows only. Figure 8a depicts simulation results for which the optimized controls are applied *only to one specific row*, with the thrust coefficient in all other rows kept at the reference value of $\widehat{C}'_T = 2$. From the figure it can be seen that only for the controls of the first row (R1) this results in a significant power increase in rows 2 and 3. This indicates that the optimal controls, as generated by the optimization at the wind-farm level, react

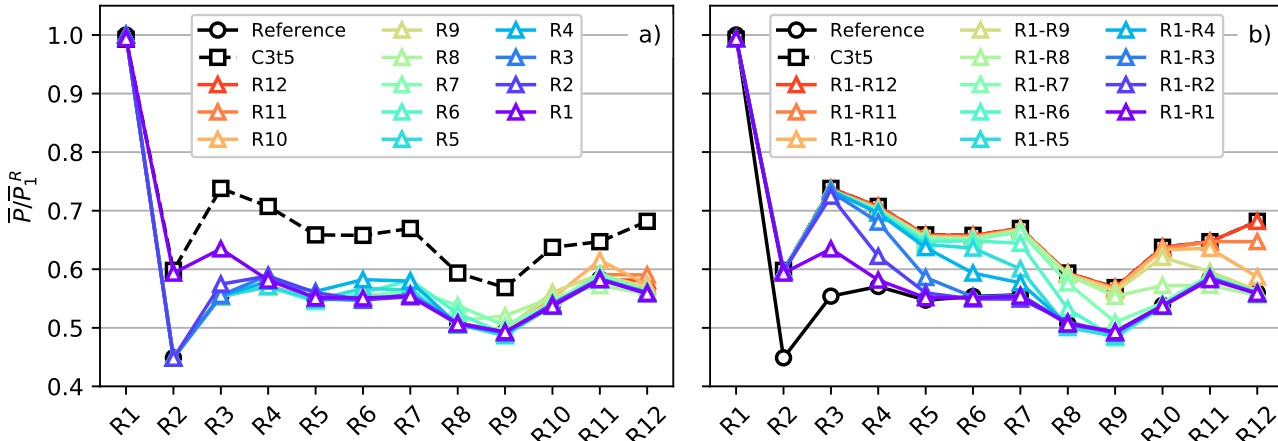

**Figure 8.** Normalized row-averaged power extraction for the reference case, optimal control case C3t5, and subset control cases up to optimization window 3. *a)* Subset control cases with optimal controls applied *only* in one specific row. *b)* Subset control cases with optimal controls applied *for all rows up to* specific row.

to the precise flow conditions caused by upstream control actions, and can hence only be applied independently for the first row, which has no upstream dependence on other controls.

Figure 8b shows results from simulations in which the controls are applied *for all rows up to a certain row*, i.e. R1–R3 indicates the application of optimized controls generated by case C3t5 to rows 1, 2, and 3. An interesting observation from this figure is that , for any row $i$ except the last one, the power potential as observed in case C3t5 is almost fully recovered by only applying the optimal controls up to row $i - 1$. This suggests that self-optimization is very limited: the optimal controls for a given turbine are designed to create favorable flow conditions in the downstream rows instead of increasing local power. Furthermore, although the discussion in the previous paragraph has shown that downstream controls are optimized with the upstream actions in mind, the converse is not true: upstream control actions do not require a specific downstream response in order to increase power in that downstream row.

### 3.3 Optimization of single active turbine rows

The previous section has shown that, based on the full-farm optimization case, the first-row controls can be applied independently from other turbine controls, whereas this does not work for the downstream rows. To further quantify the potential for increasing wind-farm power in each row of turbines, the current section considers a set of additional optimal control cases in which *only a single active row is optimized*, with all other rows remaining passive. Furthermore, by comparing the optimized controls of these cases with the full-farm optimization case C3t5, the degree of cooperation between turbines can be assessed. Note that the current single-row optimal control is not equivalent to greedy control: the optimizer still aims to increase aggregate farm power by taking into account wake interactions with downstream turbines. Furthermore, in contrast to the single-row

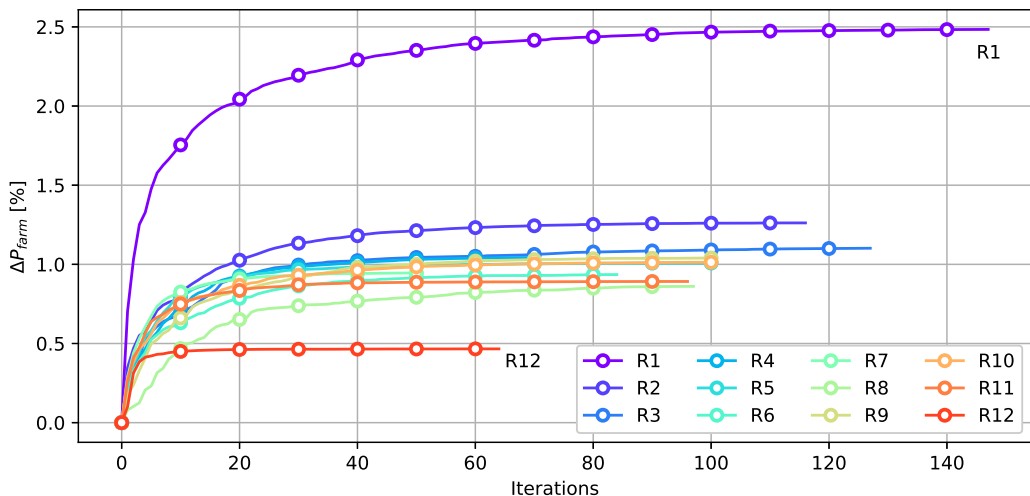

**Figure 9.** Increase in wind-farm power extraction for the first (and only) timewindow of the optimal control cases in which only a single row is optimized.

control simulations from the previous section (i.e. in Fig. 8a), the current optimizations will yield controls that are *explicitly designed* to increase power given that all other rows are passive. To limit computational costs, the additional optimizations are only performed for a single time window.

Figure 9 shows the relative increase in wind-farm power extraction for each of the twelve individually optimized control

cases. The optimization is run until the continuous adjoint gradient accuracy prevents further progress in the optimization. Upon interpreting the actual values from the figure, it is important to note that the reported power gain covers the full optimization horizon $T$, and is hence affected by finite-horizon effects. Furthermore, the first window of an optimal control simulation, as considered here, contains an initial dead zone corresponding to the wake advection lag before upstream turbines start influencing their downstream neighbors. This tends to reduce gains compared to later time windows. Nevertheless, the relative

order of the different cases still provides information that can be generalized to full optimal control studies with multiple windows and longer time horizons.

The figure shows that the first row (R1) holds by far the most promise for optimizing wind-farm power. This is not surprising as R1 produces the most power of all rows, and typically leaves the deepest wakes, causing second-row turbines to perform poorly in aligned wind-farm layouts (see, e.g., Porté-Agel et al., 2013; Nilsson et al., 2015; and Stevens et al., 2016). At the

other end of the spectrum, the last row (R12) is the least useful. The intermediate rows (R2–R11) lie closer together, with the general trend being that the potential is somewhat decreased with downstream distance into the wind farm, although this decrease is not monotonous.

Figure 10 illustrates the row-wise relative power increase matrix for each of the single-row optimization cases. The figure indicates that, for each of the optimization cases, the largest power increase is observed in the first row downstream of the active

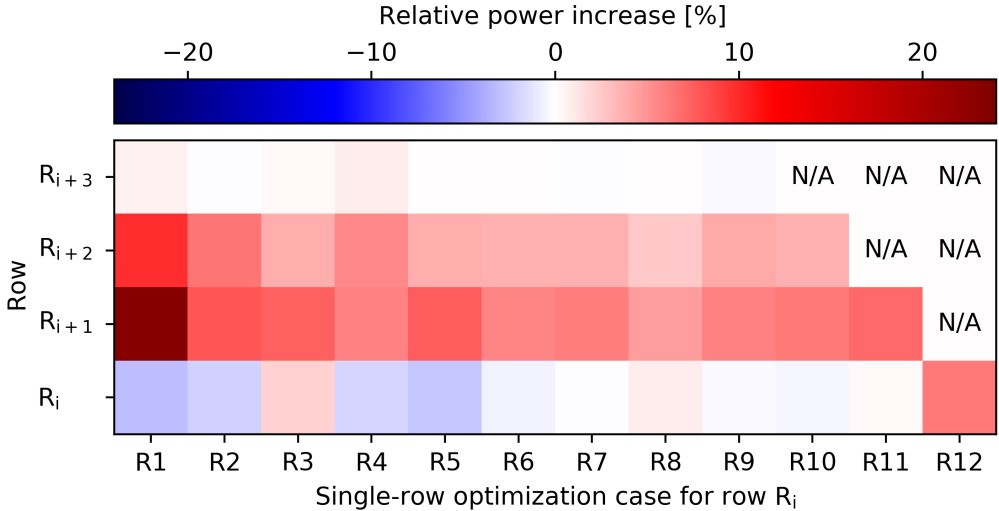

**Figure 10.** Relative power increase matrix for downstream rows in the single-row optimization case for row $R_i$, indicated in the horizontal axis. N/A indicates non-existing downstream rows. Finite-horizon effects are eliminated by only reporting power increase up until $t = T_A$ for the active row $R_i$.

turbine (i.e. $R_{i+1}$), and that the influence on row $R_{i+3}$ is limited. This is explained by the fact that the finite optimization horizon used in MM17 (i.e. $T = 240$ s) allows for more interactions with directly neighboring turbines than with those located further downstream. Furthermore, except for the optimal control case of the last row (R12), self-optimization is virtually non-existent: power gains are achieved by modifying the flow to yield more favorable conditions for downstream rows.

**3.4  Modification of thrust coefficient signals**

The observations from previous sections illustrate that, at least to some degree, the optimized thrust coefficients are tuned to local flow conditions. In the current section, the possibility of whether the coefficients contain traits that are independent of flow conditions is investigated. To this end, the optimized thrust coefficients are modified in such a way that correlations between them and specific flow events are eliminated. This is done in two independent test cases.

In the first case, the controls, which were specifically generated for selected turbines, are reassigned to other turbines by randomly swapping the control sets of different turbine columns. In doing so, each turbine will receive controls that were specifically designed for another turbine in the same row. To avoid erroneous conclusions based on coincidence, the column swap is performed in 2 random independent ways. The variability of flow conditions for different columns can be qualitatively observed in Figure 2. To further strengthen the hypothesis of the current experiment, we verified that the correlation between

flow conditions in different columns is small, i.e. with an average Pearson correlation coefficient of 0.12 between columns for the incoming velocity fluctuations $6D$ upstream of the first row. Row-averaged power for these cases is shown in Fig. 11a.

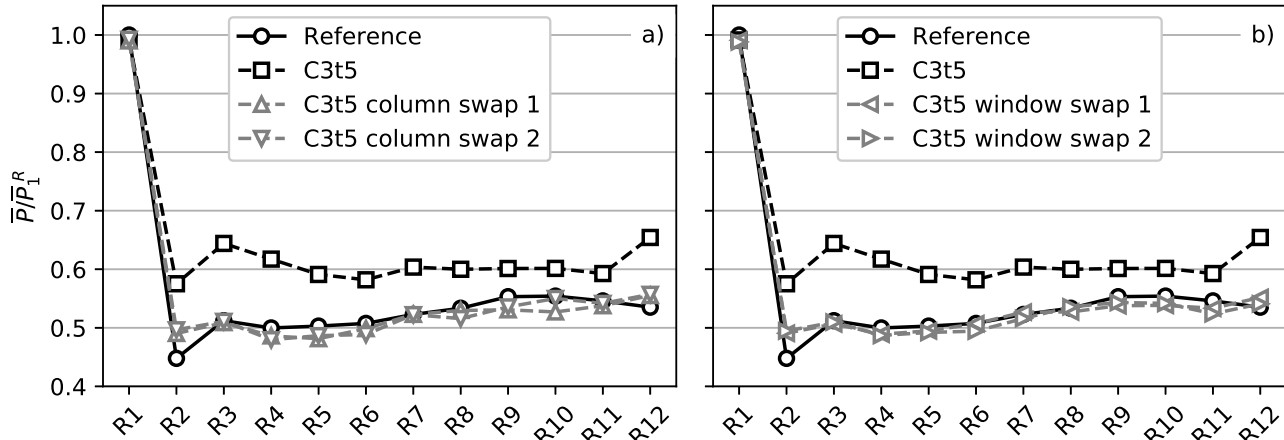

**Figure 11.** Normalized row-averaged power extraction for the reference case, optimal control case C3t5, and modified control cases. *a)* Modified control cases with controls swapped between wind-turbine columns. *b)* Modified control cases with controls swapped randomly by time window.

In the second case, controls remain assigned to their original turbines, but are shuffled in time by randomly swapping optimal controls generated for different control windows. In this way, the spectral thrust characteristics for timescales smaller that the control horizon $T_A = 120$ s remain unchanged, whereas the time synchronization of control actions to specific flow events is eliminated. Similar to the first case, this is done in 2 random ways, and the limited correlation between velocity fluctuations in different time windows was quantified at 0.07. Figure 11b illustrates the row-averaged power for these cases.

The figure shows similar behavior for each of the modified control cases: only in the second row (R2), a consistent (though small) increase in power extraction can be observed. This suggests the presence of flow-invariant features in the control signals of the first row. Note however that the full power gain in the second row is only partially attained, indicating that also the first turbine row reacts to the specific flow conditions.

## 3.5 Discussion

The observations and experiments from previous sections have revealed information that increases the understanding the optimized thrust coefficients, and can be used as a starting point towards designing practical wind-farm controllers that do not require computationally expensive LES-based optimal control simulations.

A first conclusion is that wind turbines can be classified into three distinct categories, based on their position within the farm: first-row turbines, last-row turbines, and intermediate turbines. The most salient behavior can be found in the first-row turbines (R1). It was shown that these turbines exhibit the largest variability in thrust forces and hold the greatest potential for power optimization. Furthermore, they are not influenced by upstream turbine control action, and are the only turbines that retain part of the power gains after eliminating possible correlation between controls and specific flow events. The characteristics

of the last row turbines (R12) also stand out from the rest due to the fact that, by definition, the last row has no downstream turbines and hence holds no further potential for coordinated control. The remaining intermediate rows (R2–R11) have similar spectral thrust characteristics and potential for power increase, situated in between but clearly separated from first- and last-row turbines. Further, it is worth noting that the behavior and analysis of control actions in these turbines is most complex: not only do they influence downstream turbines, they in turn are dependent on controls of upstream turbines.

A second conclusion is that, whether or not the wind-farm is controlled with the possibility of active response and cooperation between turbines, the resulting power and thrust characteristics are very similar. It was shown that self-optimization is very limited, and that, for any row $i$, the full potential in power increase is virtually attained by applying controls only for the upstream turbines up until row $i-1$. These observations strongly suggest that the optimized thrust coefficients are designed in a parabolic manner, i.e. with a unidirectional propagation of control information from the first row to the last, and very little upstream influence of downstream turbine actions. With this in mind, the following section of this paper will focus on the first and most promising link of the control chain: the turbines situated in the first row of the wind farm.

## 4   First-row turbine behavior

The current section further focuses on the analysis of the first row turbines. First, a qualitative analysis of the instantaneous flow field is performed in Sect. 4.1, resulting in the observation of vortex rings being shed from first-row turbines. Thereafter, this mechanism is mimicked by imposing sinusoidally varying thrust coefficients in these turbines in Sect. 4.2, with the aim of increasing power through similar mechanisms as in the computationally expensive optimal control cases.

### 4.1   Flow field visualization

Figure 12 shows snapshots of the vorticity and velocity fields at t = 300 s for the reference case (left) and the optimal control case C3t5 (right). Figures 12a,b show isosurfaces of vorticity magnitude, colored by streamwise velocity $\widetilde{u}_x$. Figure 12a shows that, in the reference case, the first-row turbines shed relatively stable vortex sheets that demarcate the wake from the freestream flow. The sheets destabilize and break up as they are advected downstream, resulting in complex three-dimensional vortical structures. Furthermore, as also shown in Fig. 12c, wake mixing is limited, and downstream turbines experience reduced velocities. In contrast, the optimized case shows coherent vortex rings being shed from the first-row turbine. As indicated by the black arrows in Fig. 12b, the locations of the rings in the controlled case coincide with naturally occurring bulges in the vortex sheet of the reference case: the controlled turbines further destabilize the sheet by well-timed temporal variations in its thrust coefficient. Figure 12c shows that this results in smaller velocity deficits in the wake region. Note that, downstream of the second turbine, the vorticity field becomes much more complex and differences in the flow fields are less coherent.

The observed shedding of ring vortices seems to occur at specific flow-synchronized times to exploit natural instabilities in the original vortex sheets. Therefore, the remainder of this paper will attempt to accomplish the same effect by simple sinusoidal thrust variations.

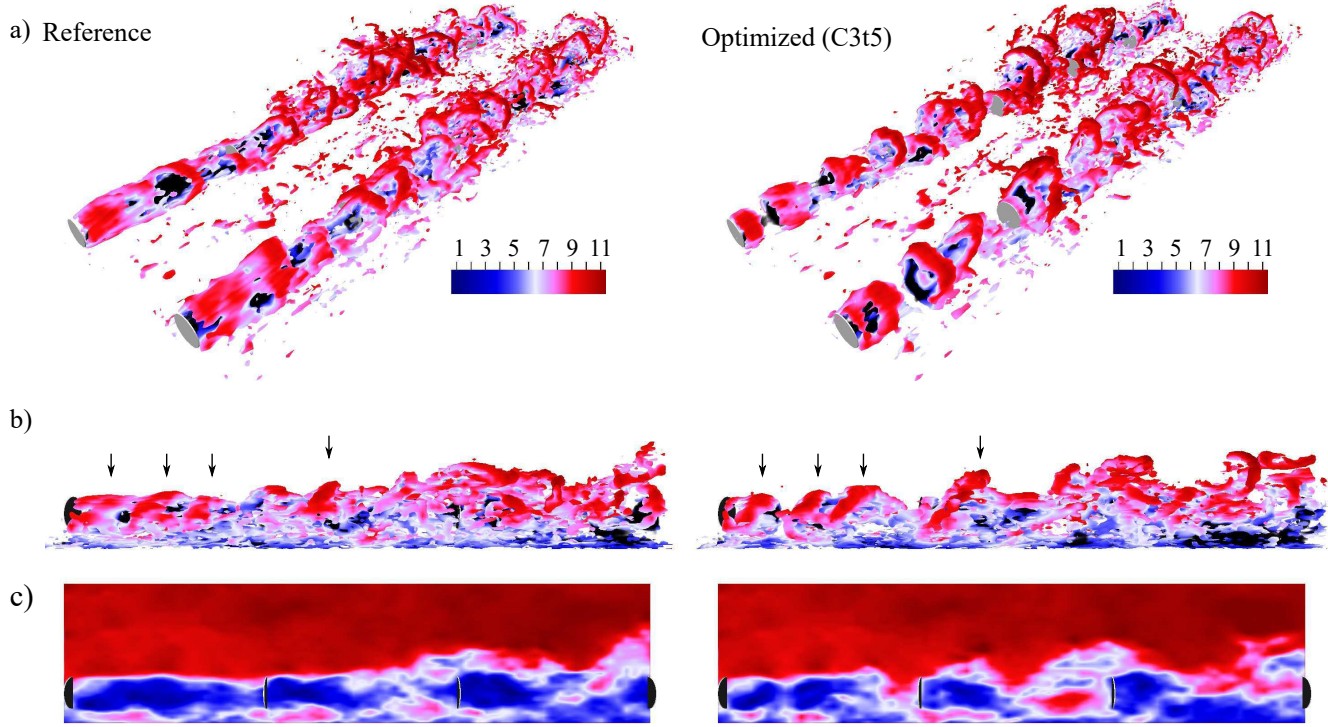

**Figure 12.** Instantaneous snapshots at t = 300 s of portion of wind-farm flow field for the reference (left) and optimized (right) case. *a,b)* Isosurface of vorticity magnitude, colored by streamwise velocity $\widetilde{u}_x$. Deep wake regions (with $\widetilde{u}_x < 4.5$ m s$^{-1}$) are rendered in black. Black arrows indicate the naturally occuring unstable bulges in the reference case, and accompanying vortex rings in the optimized case. *c)* Contours of streamwise velocity $\widetilde{u}_x$. Coloring is in units of m s$^{-1}$. Wind turbines are represented as gray disks.

## 4.2 Sinusoidal thrust variations

The aim of the current section is to mimic the quasi-periodic shedding of vortex rings by upstream turbines as observed above through the use of simple periodic variations in the thrust coefficient. Instead of optimizing a high-dimensional control signal that can evolve freely in time as in MM17, we impose a sinusoidal perturbation on the Betz-optimal coefficient $\widehat{C}'_T = 2$, parametrized by its amplitude $A$, and its frequency in the form of a non-dimensional Strouhal number $St = fD/U_\infty$, with $f$ the dimensional frequency, $D$ the turbine diameter, and $U_\infty$ the unperturbed time-averaged upstream velocity:

$$\widehat{C}'_T(t) = 2 + A\sin\left(2\pi St \frac{tU_\infty}{D}\right). \tag{6}$$

### 4.2.1 Parameter sweep

Instead of optimizing $A$ and $St$ using a similar gradient-based optimization setup as in MM17, we perform a parameter sweep to find optimal parameter combinations. The reason for this is that we would need a rather long optimization horizon $T$ to find

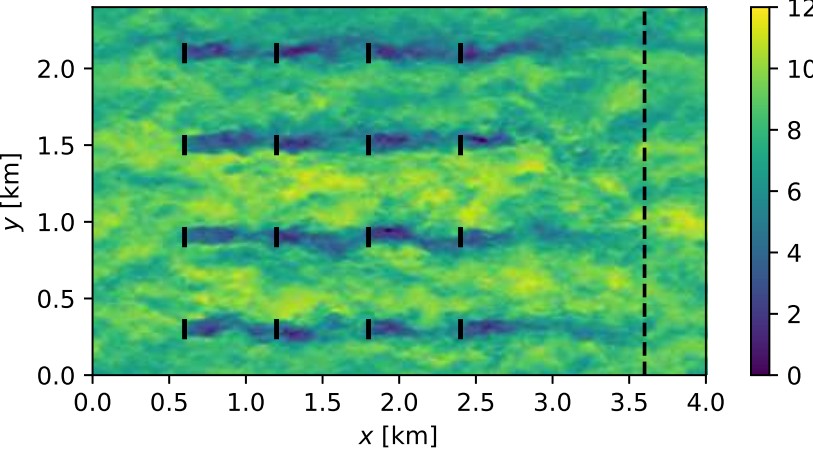

**Figure 13.** Reduced $4 \times 4$ wind-farm simulation setup for sinusoidal variation parameter study showing instantaneous contours of streamwise velocity $\widetilde{u}_x$. Coloring is in units of m s$^{-1}$. The dashed line indicates the start of the fringe region for imposition of unwaked inflow conditions.

a robust parameter combination that is independent of specific flow realizations. Unfortunately, the chaotic nature of turbulent flow fields makes long-time optimization using adjoint LES practically infeasible to date (see, e.g. Wang et al., 2014). However, the fact that we have only two parameters renders a parameter sweep computationally feasible. The sweep is performed for a reduced-size wind-farm LES, as illustrated in Fig. 13. The farm consists of $4 \times 4$ turbines in an aligned layout with $S = 6D$

in both streamwise and spanwise directions, geometrically equivalent to the optimally controlled wind farm in MM17. The simulation is performed on a domain of $4 \times 2.4 \times 1 \,\mathrm{km}^3$ with a simulation grid of $192 \times 256 \times 144$ gridpoints. A wall roughness length $z_0 = 10^{-1}$ m is used. A set of wind-farm flow simulations is advanced in time by 30 minutes, during which the front row is controlled using a sinusoidally varying thrust coefficient $\widehat{C}'_T$, as defined in Eq. (6). Within this set, the amplitude $A$ is varied between 0.5 and 2, with increments of 0.5. Furthermore, the Strouhal number $St$ is varied between 0.05 and 0.6, with

increments of 0.05. In total, this leads to 48 LES cases within the set.

    Figure 14 illustrates the power extraction for all cases considered. Figure 14a illustrates the relative power gains over the reference case. From the figure it can be seen that there is a well-defined range of values for $A$ and $St$ for which wind-farm power can be increased substantially through upstream sinusoidal thrust variations, with a maximal power increase of $\approx 5\%$ at $(St^\star, A^\star) = (0.25, 1.5)$. For instance, a Strouhal number $St = 0.25$ corresponds here to a sine wave period of $\approx 50$ s for a

turbine with diameter $D = 100$ m and a freestream velocity $U_\infty = 8.5$ m s$^{-1}$. For instance, considering the NREL 5MW blade profiles, the maximum thrust coefficient of 3.5 can be attained by slightly changing the rotor design, e.g. using a 50% increase in blade chord length and an operational tip speed ratio 25% higher than the original design value (see Goit and Meyers, 2015, Appendix A). Furthermore, given such redesign, dynamic reductions from this value could be realized through blade pitch control, for which actuation rates in the order of $10°/$s are possible (see, e.g., Jonkman et al., 2009). Figure 14b illustrates

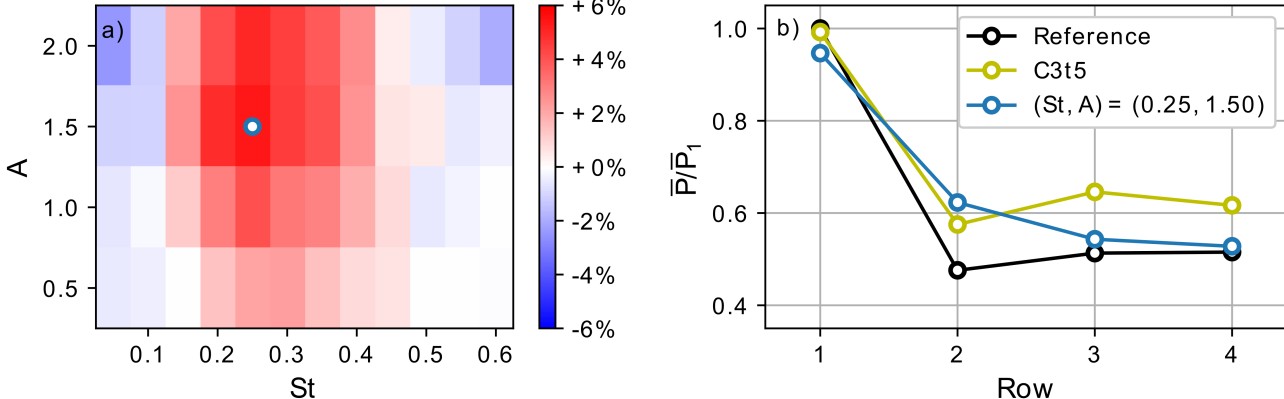

**Figure 14.** Power extraction of baseline sinusoidal thrust case ($S = 6D$, $z_0 = 10^{-1}$ m). *a)* Relative gain in mean wind-farm power extraction over reference case as a function of sine amplitude $A$ and frequency $St$. *b)* Row-averaged mean power extraction for the best sinusoidal case, the optimal control case C3t5 from MM17, and the reference case, normalized by first-row reference power.

normalized power extraction by row for the reference case, the best sinusoidal case, and the first four rows of the optimal control case C3t5 from MM17. The figure shows that the power gain in the sinusoidal cases originates mostly from the second row, and that power in the first row is decreased by approximately 5%. In contrast, optimal control case C3t5, in which all rows are active, also increases power in rows 3 and 4, and reduces first-row power by only 1%.

Figure 15 illustrates instantaneous vorticity and axial velocities for a set of wind turbines of the aforementioned reference case (a), the best sinusoidal case with $(St, A) = (St^\star, A^\star) = (0.25, 1.5)$ (b), and a sinusoidal case which does not lead to increased power extraction with $(St, A) = (0.6, 2)$ (c). The figure illustrates that sinusoidal variations in the first-row thrust coefficient indeed cause periodic shedding of vortex rings. Figure 15b shows that, at the optimal frequency, this leads to increased wake mixing, providing the second-row turbine with a higher incoming velocity. In contrast, Fig. 15c shows that,

even though higher frequency thrust oscillations also result in periodic shedding of vortex rings, this does not automatically lead to more favorable flow conditions for downstream turbines. Therefore, it can be concluded that a correct timing and spacing of vortex rings is essential for increased wake mixing.

In order to assess whether the same strategy can be used in the downstream turbines as well, Fig. 16 illustrates the results from an identical parameter sweep as discussed above, except that here the second turbine row is controlled using a sinusoidal

thrust coefficient. Figure 16a indicates that sinusoidal actuation of the second row invariably leads to losses in wind-farm power. Figure 16b shows that, for the optimal combination of parameters of $(St, A) = (0.25, 1.5)$ as reported for first-row actuation above, the minor power increase in the third row does not compensate for additional losses in the second and fourth row. This shows that the proposed simple control strategy does not work when applied to waked turbines, and that more elaborate control strategies are required to harness the gains achieved by the optimal control simulation in the downstream regions of the farm.

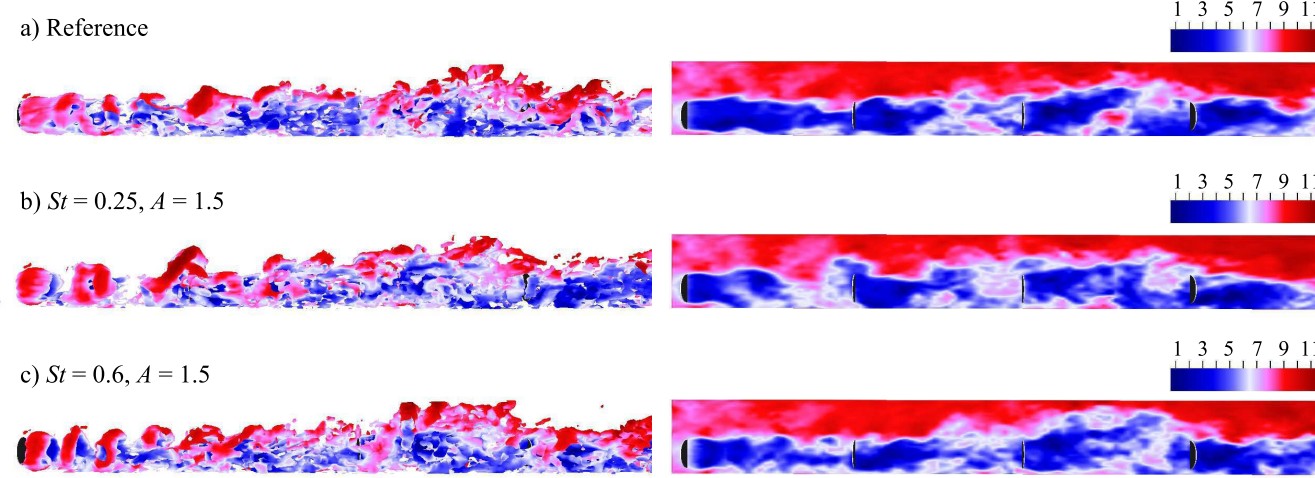

**Figure 15.** Snapshots of wind-farm flow fields at t = 1 800 s. *Left:* Isocontours of vorticity, colored by streamwise velocity. *Right:* Contours of streamwise velocity. *a)* Reference. *b)* Best sinusoidal case, with $(St, A) = (St^\star, A^\star) = (0.25, 1.5)$. *c)* Sinusoidal case with $(St, A) = (0.6, 2)$

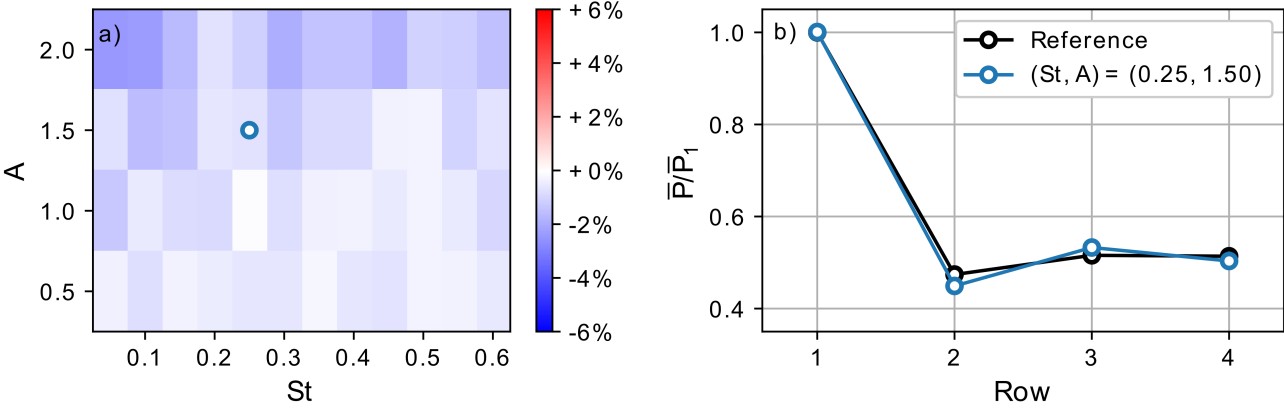

**Figure 16.** Power extraction of second-row sinusoidal thrust case . *a)* Relative gain in mean wind-farm power extraction over reference case as a function of sine amplitude $A$ and frequency $St$. *b)* Row-averaged mean power extraction for the sinusoidal case with the optimal parameters as for row 1 sinusoidal thrust, and the reference case, normalized by first-row reference power.

It is interesting to note that, even though the current parameter sweep is performed using different initial and inlet conditions than those applied in MM17, the optimal frequency of sinusoidal variations in $\widehat{C}'_T$ at $St^\star = fD/U_\infty = 0.25$ corresponds to the location of the peak in the first-row thrust coefficient spectrum of C3t5 in Fig. 7. In the following paragraphs, the robustness of the best parameter pair for first-row thrust variations, i.e. $(St, A) = (0.25, 1.5)$, is investigated with the aim of assessing the general applicability of this control strategy. To this end, similar parameter sweeps are performed for cases with varying turbine spacings and turbulence intensities.

### 4.2.2 Robustness with respect to turbine spacing and turbulence intensity

Figure 17 shows the power extraction resulting from two parameter sweeps with streamwise turbine spacings of $5D$ and $7D$ respectively. Results are promising: for the given cases, $St^\star$ and $A^\star$ do not depend on streamwise turbine spacing. Furthermore, even in the $7D$ spacing case (Fig. 17c-d), which naturally features lower overall power losses in downstream rows, power extraction in the second row can be significantly increased through sinusoidal variations in the first-row thrust coefficient.

Figure 18 depicts the power extraction results from a parameter sweep with the same wind-farm layout as in the baseline case, but with a tenfold increase in roughness length, i.e. $z_0 = 1$ m. This results in a turbulence intensity of approximately 16%, compared to 10% in the baseline case. Again, the best parameter combination of $(St^\star, A^\star) = (0.25, 1.5)$ remains unchanged. Further, even for this higher turbulence case, in which downstream losses are lower due to naturally better wake mixing, power is increased in the second row, leading to a relative gain in wind-farm power of around 2%.

As evidenced above, periodic sinusoidal variations of first-row thrust coefficients substantially increase power extraction in the second row, resulting in a net gain in total power for the considered $4 \times 4$ wind farm. Moreover, different simulation sets indicate that, at least for the range considered here, the best values for Strouhal number and amplitude of these variations, i.e. $(St^\star, A^\star) = (0.25, 1.5)$, are robust with respect to turbine spacing and turbulence intensity.

### 4.2.3 Full-scale wind-farm LES

In the remainder of this section, the sinusoidal variation strategy will be tested in a full-scale wind farm LES, corresponding to the full $12 \times 6$ aligned wind farm of MM17. Simulations are performed for a reduced range of amplitudes and Strouhal numbers, corresponding to the most favorable region identified in the parameter sweeps above. In order to increase statistical convergence, the time horizon for each simulation is extended to a physical time of 10 hours.

Figure 19 shows the power extraction of the full-scale LES. Figure 19a shows the relative power gains over the reference case for the full wind farm. It can be seen that the total power gain or loss is below 0.5% for each of the sinusoidal control cases. Figure 19b shows the row-wise power extraction for the reference case, the sinusoidal thrust case with $(St, A) = (0.25, 1.5)$, and the optimal control case C3t5. It is shown that, although the second row of the sinusoidal thrust case achieves similar power gains as those observed above, from the fifth row onwards power is slightly reduced in the sinusoidal case.

The top panel of Fig. 20 shows cross sections of time-averaged axial velocities $\widetilde{U}_x$ at the rotor disk locations for the reference case. Further, the middle and bottom panel illustrate deviations from the reference velocity for the $(St^\star, A^\star)$ sinusoidal case and the optimal control case C3t5 respectively. The figure shows that both controlled cases show similar characteristics at the

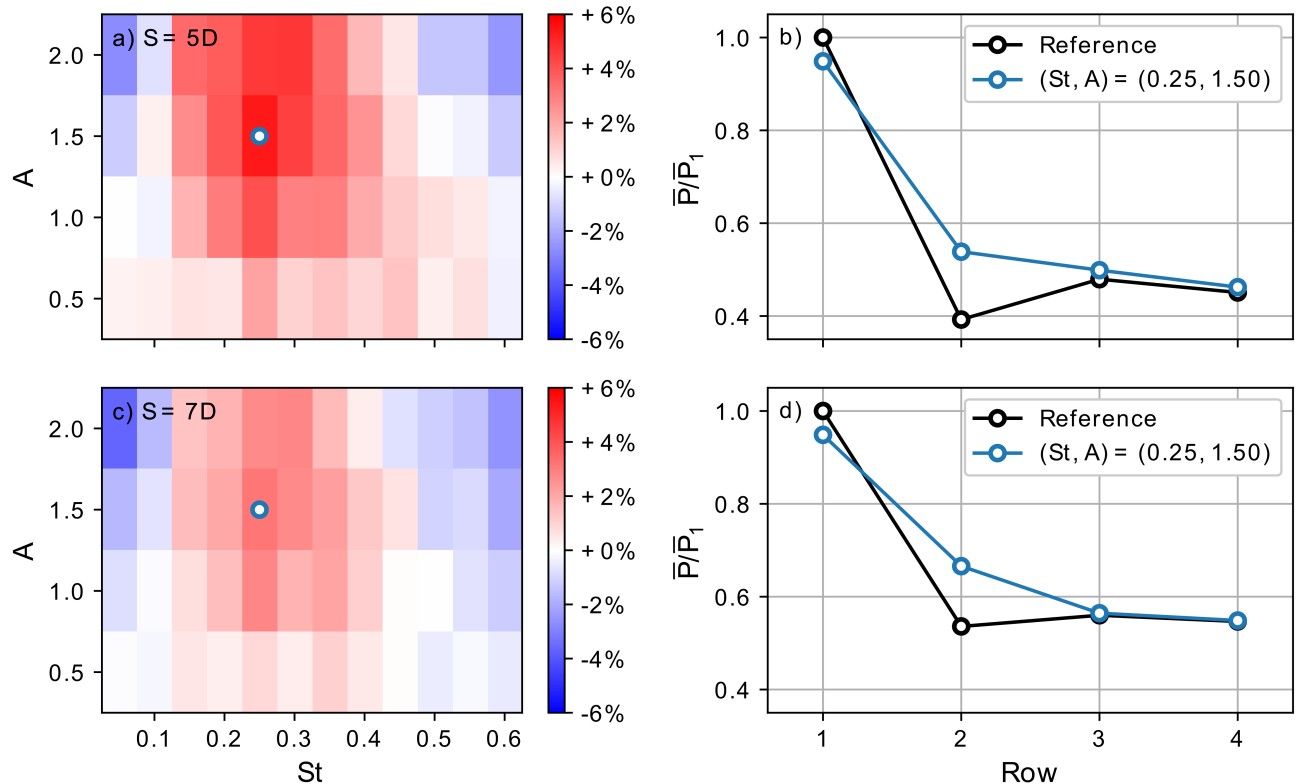

**Figure 17.** Power extraction of sinusoidal thrust cases with varying streamwise spacing. *Top (a,b)*: Decreased spacing at $S = 5D$. *Bottom (c,d)*: Increased spacing at $S = 7D$. *Left (a,c)*: Relative gain in mean wind-farm power extraction over reference case as a function of sine amplitude $A$ and frequency $St$. *Right (b,d)*: Row-averaged mean power extraction for the best sinusoidal case and the reference case, normalized by first-row reference power.

second turbine row, with an increased axial velocity at the rotor disk, accompanied by decreased velocities above and below. Downstream, it can be seen that the passive turbines of the sinusoidal case fail to retain increased velocities at the rotor disks, instead resulting in slightly lower disk velocities starting from the fifth row. In contrast, case C3t5, in which all turbines are actively controlled, succeeds to attain similar cross section characteristics with higher rotor velocities in the downstream as well. Note also that, for the fifth row, the disk velocity is slightly lower for the sinusoidal control case than for the reference case, consistent with the decreased power extraction observed in Fig. 19. This can be explained by the fact that first-row control actions cause enhanced entrainment of momentum from the internal boundary layer above the turbine canopy that would otherwise be entrained by natural turbulent mixing in passive downstream rows. In consequence, lesser entrainment occurs for downstream rows, resulting in a slight decrease in disk velocities from the fifth row onwards.

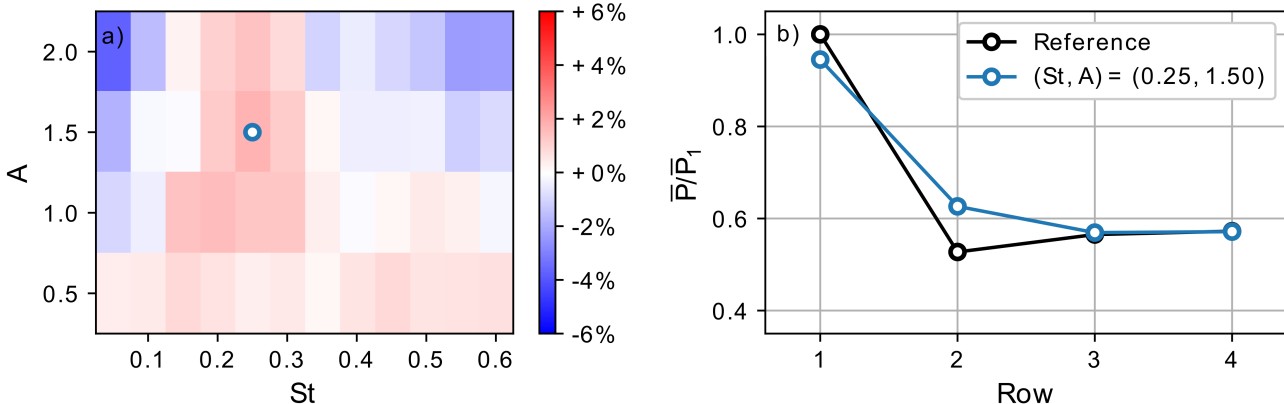

**Figure 18.** Power extraction of sinusoidal thrust case with increased wall roughness $z_0 = 1$ m. *a)* Relative gain in mean wind-farm power extraction over reference case as a function of sine amplitude $A$ and frequency $St$. *b)* Row-averaged mean power extraction for the best sinusoidal case and the reference case, normalized by first-row reference power.

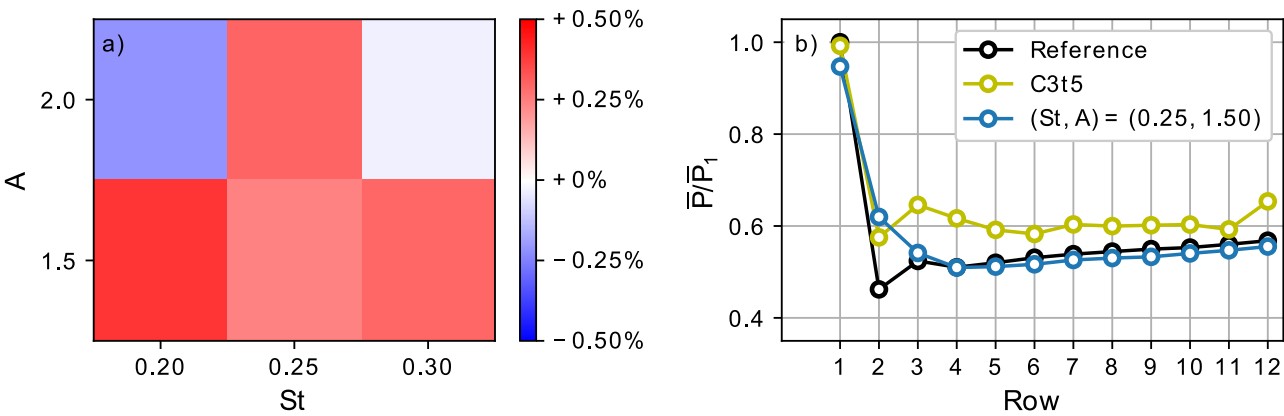

**Figure 19.** Power extraction of full-scale sinusoidal thrust case ($S = 6D$, $z_0 = 10^{-1}$ m). *a)* Relative gain in mean wind-farm power extraction over reference case as a function of sine amplitude $A$ and frequency $St$. *b)* Row-averaged mean power extraction for the sinusoidal case with parameters from previous sections, the optimal control case C3t5 from MM17, and the reference case, normalized by first-row reference power.

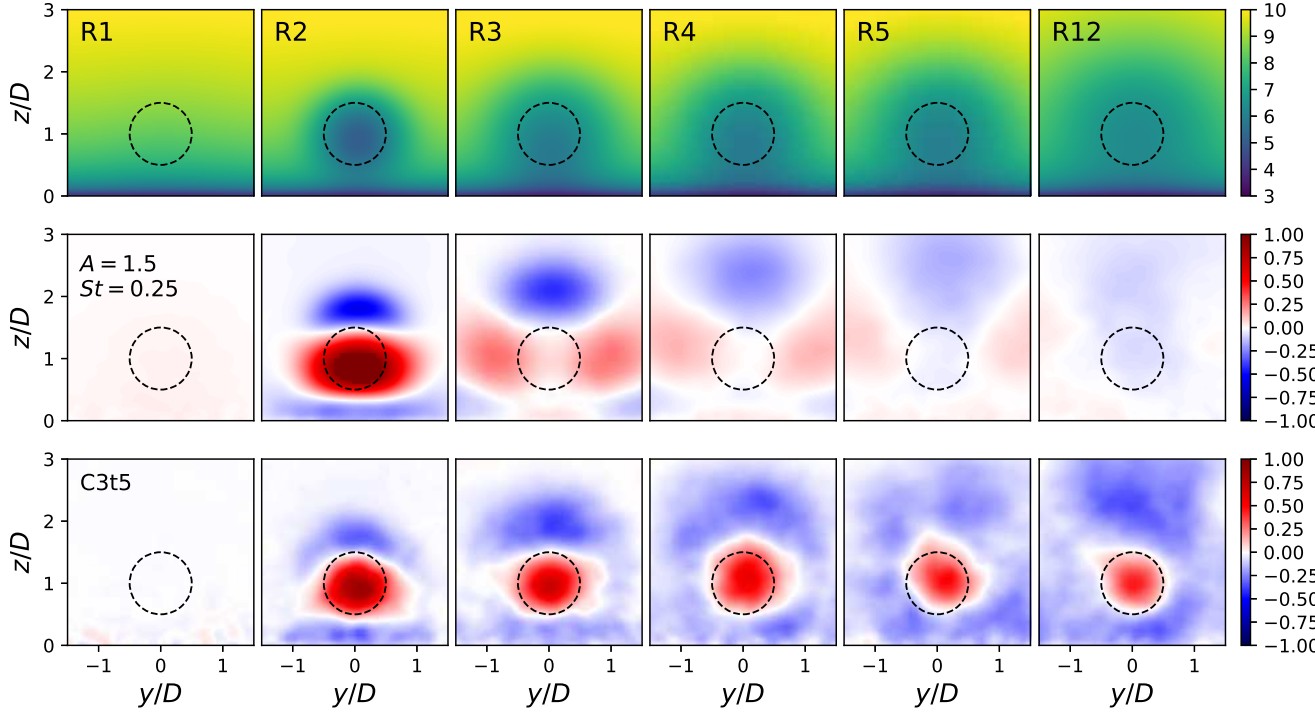

**Figure 20.** Cross section of time-averaged axial velocity $\widetilde{U}_x$ at rotor locations in rows 1, 2, 3, 4, 5, and 12. *Top:* Reference case. *Middle:* Difference between best sinusoidal perturbation case (with $A = 1.5$, $St = 0.25$) and reference case. *Bottom:* Difference between optimal control case C3t5 and reference case. Coloring is in units of m s$^{-1}$.

As shown throughout the current section, a qualitative analysis of instantaneous flow features in the optimal induction control wind farm from MM17 has led to the identification of a sinusoidal thrust control strategy for first-row turbines, resulting in increased power extraction in the second row. However, important comments should be made. First, sustained sinusoidal thrust variations with a large amplitude could contribute significantly to turbine fatigue loading of the first-row turbines. Furthermore, partial wake alleviation and unsteady passing of abovementioned vortex rings could also increase fatigue loading in downstream rows. Hence, structural aspects should be taken into account upon evaluating the practical viability of the approach. Second, even though experiments have shown that, for practically relevant tip speed ratios, wind turbines shed vortices in a similar way as disk-like bluff bodies (Medici and Alfredsson, 2006), the current behavior could still be an artifact of the relatively simple ADM used throughout this study. Further verification using higher fidelity wind turbine models, such as actuator line models, and wind-tunnel testing is hence necessary.

## 5   Intermediate-row turbine behavior

The current section discusses the intermediate rows. It was shown that, without active participation in these rows, upstream gains are lost in downstream rows, and only full optimal control succeeds in achieving significant gains in downstream rows as well (see Figs. 19, 20). It was already mentioned that the analysis and behavior of turbines within intermediate rows is more complex than in the first row: they aim to influence the flow to the benefit of downstream rows but are also dependent on the actions of upstream rows. The remainder of the current section aims to illustrate the additional difficulty for power increase in downstream rows, and speculates on possible future paths for the identification of simplified control strategies as found for the first row.

First, as shown in the top panels of Fig. 20, even in the uncontrolled case, the kinetic energy of the flow in the vicinity of the turbine rotor is depleted more and more in the downstream rows. This complicates control strategies for these rows as the opportunity for increased mixing with high energy flow is decreased. Furthermore, intermediate turbines are subjected to increased turbulence levels and more complex vorticity dynamics, as illustrated in Fig. 12. This could explain why sinusoidal thrust control did not lead to increased power when applied to the second row: whereas the first row produces increased mixing by destabilizing relatively stable vortex sheets into vortex rings, the second row is already continuously immersed in complex vorticity patterns for which this simple approach does not work. However, note that for instance R7 in Fig. 5 also seems to show quasi-periodic sinusoidal variations in $\widehat{C}_T'$ at a time period of approximately 50 s. This is an indication that, also for intermediate rows, vortex ring shedding could amount to part of the power increase observed in the optimal control simulations, albeit at specific moments in time, synchronized with the local flow conditions.

Second, it is important to note that the vortex ring shedding mechanism constitutes only part of the power increase caused by the first row. Figure 8 illustrates that the first-row optimized thrust coefficient also results in a significant power increase in the third row, which is not observed using the sinusoidal thrust strategy. Furthermore, the analysis of the modified control cases in Fig. 11 proves that also the first-row controls are partially synchronized with the flow. This shows that other mechanisms, dependent on specific flow events for increasing wind-farm power, are at play as well. Even though the application of regression algorithms in an attempt to link turbine actions to low-dimensional flow measurements (e.g. local velocity, shear and kinetic energy) has been unsuccessful thus far, similar analysis based upon more complex flow features (e.g. vorticity structures, high-speed turbulent streaks, or downdrafts) might be more promising. This requires further optimal control simulations over an extended time, as the total control time horizon of 30 minutes in the current dataset is insufficient for robust statistics in this kind of analysis. This is an important remaining challenge to be addressed in future research.

## 6   Conclusions

The current paper provided an analysis of the thrust coefficient control characteristics for the C3t5 optimal control case featured in Munters and Meyers (2017).

Analysis of the thrust coefficients and numerical experiments have shown a clear distinction between first-row turbines, last-row turbines, and intermediate turbines. Furthermore, observations strongly suggest that the optimization works in a uni-

directional way: upstream turbines influence the flow field resulting in favorable conditions for their downstream neighbors, yet information on the possibility of active response and cooperation in the latter has no influence on upstream control actions.

Qualitative analysis of instantaneous flow fields led to the observation of quasi-periodic shedding of vortex rings from first-row turbines in the optimal control case. This flow feature was succesfully mimicked using simple sinusoidal thrust actuation of the first row. The best parameter set for these sinusoidal variations proved robust to both wind-turbine spacing and turbulence intensity, with an amplitude $A^\star = 1.5$ and a non-dimensional frequency at $St^\star = 0.25$. Interestingly, this frequency corresponds to the peak at $St = 0.2\ldots0.3$ observed in the first-row thrust coefficient spectra of the optimal control case. Although the first-row sinusoidal control led to a robust increase in total power for a reduced-size $4 \times 4$ wind farm, a full-scale test indicated that downstream turbine activity is required to obtain increased power at larger farm scales. It was also shown that the simple sinusoidal strategy does not lead to increased power extraction when applied to downstream intermediate turbines. Identifying the mechanisms for power increase in these turbines hence remains an important open research question. Finally, it is important to remark that all current simulations were performed using a standard non-rotating actuator disk model without the inclusion of mechanical turbine loading. Therefore, wind tunnel testing and/or simulations with more advanced turbine models (such as the actuator line model) including assessment of turbine loading are essential to evaluate the real-life applicability of the sinusoidal thrust strategy.

*Competing interests.* The authors declare no conflict of interest.

*Acknowledgements.* The authors have received funding from the European Union's Horizon 2020 research and innovation programme under grant agreement No. 727680 (TotalControl). The authors also acknowledge funding by the European Research Council under grant agreement No. 306471 (ActiveWindFarms). The computational resources and services used in this work were provided by the VSC (Flemish Supercomputer Center), funded by the Research Foundation - Flanders (FWO) and the Flemish Government, department EWI.

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
