# Peer review of "Towards practical dynamic induction control of wind farms: analysis of optimally controlled wind-farm boundary layers and sinusoidal induction control of first-row turbines"

_Wind Energy Science, 2018_

## Referee Comment (RC1) · Anonymous Referee #1 · 6 Apr 2018

This was a good and interesting paper. The results are well presented, and the explanations are clear. The authors use an interesting approach to explore what is possible with induction-based wind farm control and find a method which can be reduced to a simple control law and increases power in LES. The authors use the results to make interesting points which might be fruitfully followed-up in their own research, or can inform similar research in wind farm control.

For example, the result that much of the value of axial control is produced by the first row alone is an interesting, and useful result, as it might imply that the job of implementing wind farm control can be limited to perimeter turbines. Perhaps this result will hold for other types of wind farm control.

The sinusoidal frequency used to encourage wake recovery, its presence in the optimal solution's spectrum, and its resilience to changes in turbine and timing is an additional useful result. Finally, the flow analysis and discussion adds to the intuition of how this, and more generally, how all wind farm controllers finally will gain power. For example the point that the flow over the intermediate turbines is lower energy then what flows over the first turbines, and so can only be less-profitably mixed down is a good point I hadn't seen made before.

The main problem confronting the approach appears to be the probable impact on turbine loads in varying the thrust to such a degree. However, the authors acknowledge this and identify this several times in the paper as a subject of future work. It would be an interesting problem to see, perhaps using a more detailed turbine model, if the required forcing of the flow can be generated in a realizable way with loads that not too severe. With low-enough frequencies this would seem at least possible, but perhaps the fact that the peak thrust is being raised in the Ct3 controllers will unavoidably raise loads.

A second issue might be the gain in power appears to be very small in the latter sections where a full wind farm is simulated (unless I misunderstand). Considering that LES simulations typically simulate one of the more promising wind directions for a wind farm, back of the envelope often suggests a gain in 10% or similar for an overall increase of AEP to be significant when all wind directions are considered. But perhaps this is a misunderstanding of section 4.2.3.

However, the paper contributes insightful analysis and presents an interesting study of how axial-based wind farm control can successfully raise power in LES, which had seen to be a challenge using static points. Further, the insights of the paper I believe are generally useful for understanding the underlying challenge of wind farm control.

[Figure]

Very interesting.

Specific comments:

One comparison that came to mind reading the paper, is that the results show very little value in optimizing the most-downstream turbines, and in general, improvement in power comes from modifications upstream, and that self-optimization is not possible. This stands in contrast to a result such as:

Ciri, Umberto, Mario Rotea, Christian Santoni, and Stefano Leonardi. "Large Eddy Simulation for an Array of Turbines with Extremum Seeking Control." In American Control Conference. Boston, MA, 2016.

Where the TSR of downstream turbines is re-optimized for wake conditions (and the upstream turbine is left as is at the end of the optimization). It would seem the difference in modeling methods and/or how turbine control is implemented yields the different results, but I believe it would be worth discussing the difference, for example around the paragraph beginning with "The figure shows that the first row (R1)..." on page 12.

Figure 3/related text: Would another way to describe Ct2 vs Ct3 be that Ct2 can only lower the thrust, while Ct3 is allowed to raise it?

Page 17: "...NREL 5MW rotor with a 50% increase in chord length..." does this imply the method is currently assuming the chord length is variable? Could this not be achieved by change in pitch angle?

---

## Referee Comment (RC2) · Anonymous Referee #2 · 16 Apr 2018

The authors submitted a well-written paper, which focuses on one of the hottest topics within wind energy applications: wind farm control. The possible power gain with induction based control strategies via further enhancing the turbulent mixing is investigated. The presented analysis of the dynamic induction control starts with further assessment of the previous study, yet the necessary background work is included properly so that the paper still reads stand-alone.

Distributing CT set-points dynamically to use the turbines as flow actuators is a novel and a very interesting idea. The control framework and the considered wind farm scale

[Figure]

LES setup is described/illustrated properly - except of the characteristics of the turbines/actuator disks considered. Would be nice to have the size of the disks explicitly noted (as they are somewhat hidden in Figure 2) to have a more clear scale of the considered wind farm. From here on, the comments are more detailed and listed following the structure of the manuscript.

- While describing the case setup in Section 2.2, the "flow advancement time", TA (also referred in Figure 1) is considered as half of the prediction horizon T. Would TA (and therefore T) be inflow dependent as the time delay (the time it takes for particles to move from the upstream to downstream turbine(s))? Have you investigated if changing T (and/or TA) has any effects on the resulting optimum CT set-points and on the power gain?

- As clearly seen in Figure 4c and 4d, there is a significant increase in turbulence further downstream. In addition to the TKE and the transport, would be nice to have the turbulence intensity TI values (as listed later on page 20, 10% for the baseline case), both for the baseline case and the maximum added TI reported - possibly somewhere around Figure 4. That again would give an indication on the applicability compared to the field values observed. Also note the typo in the caption of Figure 4: after c) all the subplots are marked to be continuously c).

- On page 10, around line 10, the argument of "upstream actions do not require a specific downstream response in order to increase power in that downstream row", which is also paraphrased in the conclusions, needs to be elaborated. This rather broad conclusion seem to oversee the probability of the curtailment of the downstream turbine where down-regulation might be inevitable for certain CT set-points assigned to downstream turbine(s) in the resulting optimization. Could be partially true for the investigated C3t5 case since there observed very limited curtailment even at the most upstream turbine (as in Figure 3b). However, also seen in Figure 8b (except of the very last row as the authors indicated), there seem to be still a difference between on the power gain at turbine R11 for the scenarios of R1-R10 and R1-R11. Narrowing
the argument to the considered case or very little to no downstream curtailment CT distributions is suggested.

- On page 13, line 13, "the presence of the flow invariant features of the control signals" needs further justification as Figure 11 would also depend on how variant the flow features are in the simulations. That should include both the spatial and temporal variance within the 30-min window. As far as the field measurements are concerned, high spatial and temporal correlations are observed. For the former, Figure 2 gives a brief idea about the wind speed range between the columns, that can be referred here. For the latter, time series or relevant temporal statistics can be presented to assess the randomness and strengthen the hypothesis.

- On page 17, around line 5, a very nice example on how to implement the optimized sinusoidal CT is presented. The practical examples can be further improved by a short discussion on the expected response time of such increases in tip speed ratio on a machine with high inertia. That would put the estimated sine wave period into perspective as well.

- For Section 4.2.3, the header "Full-scale wind farm test" is a bit misleading... Suggest to change to "Full-scale wind farm simulations (in LES)" instead.

- On Figure 19, why would the power decrease after Row #5 for the sinusoidal case?

- Page 22 around line 5, the (inevitable) discussions on loads are included. In addition to the loads on the controlled upstream turbine, Figure 20(b) indicates partial wakes on the further downstream rows of turbines. Therefore, the section should be improved by highlighting the probable increase in fatigue loading for not just turbine(s) R1 but for the downstream rows as well, possibly starting as early as R3.

On the grammatical note, the manuscript is clear and easy to follow. The only comment might be on the use of Sect. or Section; Fig. or Figure references.

Very interesting and innovative work!

---

## Author Comment (AC1) · 23 May 2018

We thank the reviewer for his/her constructive comments and are pleased with the positive assessment or our work. We have addressed the specific comments made by the reviewer as described below. We hope that the revised manuscript can now be accepted for publication.

. . . . . . . . . . . . . . . . . . . . . . . . . . . . . . . . . . . . . . . . . . . . . . . . . . . . . . . . . . . . . . . . . . . . . . . . . . . . . . . . . . . . . . . .

**1. Reviewer:** *One comparison that came to mind reading the paper, is that the results*

[Figure]

*show very little value in optimizing the most-downstream turbines, and in general, improvement in power comes from modifications upstream, and that self-optimization is not possible. This stands in contrast to a result such as:*

*Ciri, Umberto, Mario Rotea, Christian Santoni, and Stefano Leonardi. "Large Eddy Simulation for an Array of Turbines with Extremum Seeking Control." In American Control Conference. Boston, MA, 2016.*

*Where the TSR of downstream turbines is re-optimized for wake conditions (and the upstream turbine is left as is at the end of the optimization). It would seem the difference in modeling methods and/or how turbine control is implemented yields the different results, but I believe it would be worth discussing the difference, for example around the paragraph beginning with "The figure shows that the first row (R1)..." on page 12.*

**Response:** This is an interesting comment. It is correct that the difference originates from the way turbines and their controls are modeled in the current study compared to that of Ciri et al. The reference disk-based thrust coefficient $C'_T = 2$ used here actually implies intrinsic self-optimization of blade pitch and generator torque to local flow conditions, even though these actions are not resolved by our current actuator disk formulation which directly controls $C'_T$. In case these degrees of freedom would be resolved, for instance using an actuator line model as in Ciri et al., the ESC from the latter study could e.g. be used to optimize torque controller gain in order to achieve an effective $C'_T \approx 2$.

We have included the fact that the reference case already implies self-optimization in the revised manuscript as follows (p 4, line 21):

" A conventionally (greedily) controlled wind farm with steady $C'_T = 2$ was defined as a reference case. **Note that this would correspond to a farm with ideal turbines for which generator torque is being controlled dynamically to track the maximum power point at the Betz limit perfectly. In a real turbine controller this may, e.g., be achieved with the extremum seeking control proposed by Ciri et al. (2017).**

Several different optimal control cases were defined, ... ”

.......................................................................................

**2. Reviewer:** *Figure 3/related text: Would another way to describe Ct2 vs Ct3 be that Ct2 can only lower the thrust, while Ct3 is allowed to raise it?*

**Response:** Indeed, this is correct. We have mentioned this explicitly in the revised manuscript as follows (p. 4, line 26):

“ ... and the maximal thrust coefficient $C'_{T,\text{max}} = 2$ or 3, **with thrust forces that can respectively only be reduced (underinductive), or also increased (overinductive) compared to the Betz optimum at** $C'_T = 2$ (see Eq. 5). ”

.......................................................................................

**3. Reviewer:** *“... NREL 5MW rotor with a 50% increase in chord length ...” does this imply the method is currently assuming the chord length is variable? Could this not be achieved by a change in pitch angle?*

**Response:** No, the method does not imply a variable chord length.

Current turbines are designed to approach maximum $C'_T$ values around 2, corresponding to the Betz limit. Aiming to increase thrust by adapting the pitch angle would inevitably lead to severe efficiency losses due to stall on the turbine blades. Therefore, we provide an example of how an alternative turbine design (i.e. with an increased chord length and operational TSR) could attain a maximum $C'_T$ of 3.5. Given such a turbine design, achieving thrust ratings $C'_T < 3.5$ is straightforward by pitching blades towards the feather position.

We have slightly modified the statement in the revised manuscript to avoid any confusion with regard to possibly having the chord length as a control variable. (p. 17, line 25):

“ **For instance, considering the NREL 5MW blade profiles, the maximum thrust**

**coefficient of 3.5 can be attained by slightly changing the rotor design, e.g. using a 50% increase in blade chord length and an operational tip speed ratio 25% higher than the original design value (see Goit and Meyers, 2015, Appendix A). Furthermore, given such redesign, dynamic reductions from this value could be realized through blade pitch control, for which actuation rates in the order of $10°/$s are possible (see, e.g., Jonkman et al. 2009). "**

Note that this control strategy is not unique and is solely given as an indication for the technical feasibility of tracking the proposed $C_T'$ waveform. In practice, we expect that this can also be achieved through a combination of generator torque and blade pitch control. This is subject of ongoing investigation.

---

## Author Comment (AC2) · 23 May 2018

We thank reviewer 2 for reading our work and for providing detailed feedback which has improved the quality of the manuscript. We are pleased with his/her positive assessment of our research and have addressed the specific comments in the manuscript as described below. We hope that the revised manuscript can now be accepted for publication.

. . . . . . . . . . . . . . . . . . . . . . . . . . . . . . . . . . . . . . . . . . . . . . . . . . . . . . . . . . . . . . . . . . . . . . . . . . . . . . . . . . . . . . . . . . . . . . . .

**1. Reviewer:** *...LES setup is described/illustrated properly - except of the character-istics of the turbines/actuator disks considered. Would be nice to have the size of the disks explicitly noted (as they are somewhat hidden in Figure 2) to have a more clear scale of the considered wind farm.*

**Response:** Indeed, we seem to have overlooked specifying the exact turbine dimensions in Section 2.2. We have updated the manuscript as follows (p. 4, line 15):

" ... 12 rows by 6 columns. **The wind turbines have a hub height** $z_h = 100$ **m with a rotor diameter** $D = 100$ **m, and are spaced apart by** $6D$ **in both axial and transversal directions.** "

. . . . . . . . . . . . . . . . . . . . . . . . . . . . . . . . . . . . . . . . . . . . . . . . . . . . . . . . . . . . . . . . . . . . . . . . . . . . . . . . . . . . . . . . . .

**2. Reviewer:** *While describing the case setup in Section 2.2, the "flow advancement time", $T_A$ (also referred in Figure 1) is considered as half of the prediction horizon $T$. Would $T_A$ (and therefore $T$) be inflow dependent as the time delay (the time it takes for particles to move from the upstream to downstream turbine(s))? Have you investigated if changing $T$ (and/or $T_A$) has any effects on the resulting optimum $C_T$ set-points and on the power gain?*

**Response:** It is true that $T$ and $T_A$ are important parameters in the receding-horizon approach. The optimization horizon $T$ can indeed be considered inflow-dependent. In our case, it is chosen as the time it takes for the flow to pass approximately four rows of turbines. In theory, it is desirable to have $T$ cover an entire wind-farm throughflow. In practice however, control over long time horizons is complicated by the chaotic nature of turbulence accompanied by adjoint gradient inaccuracies, and we choose $T$ as long as possible without having the optimization be affected by these inaccuracies (i.e. 240 s). The sensitivity of the power gains to the time horizon has not been formally quantified, yet it is reasonable to expect that the potential for beneficial interaction between turbines that are located more than three rows apart is limited.

The choice of $T_A$ is driven by conflicting incentives of reducing computational cost ($T_A \to T$) and mitigating finite-horizon effects ($T_A \to 0$). As a rule of thumb, we have been using $T_A = T/2$ in previous work. In the MM17 paper we have also tried $T_A = T/4$ and found that the effect on power gain was limited.

Both concepts described above are discussed in more detail in the MM17 paper. To keep the description of the case setup concise in the current work, we simply add that $T = 240$ s corresponds to the time to pass four turbine rows, and further refer to MM17 for further elaboration of methodological choices. We have updated the revised manuscript as follows (p. 4, line 19):

" ... with a prediction horizon $T = 240$ s **(i.e. the time it takes for the flow to pass four rows of turbines)** and a flow ... ",

and (p. 4, line 28):

" ... and $\tau = 30$ s. **The choice of (and sensitivity to) setup parameters is further elaborated in MM17.** "

...................................................................................................

**3. Reviewer:** *As clearly seen in Figure 4c and 4d, there is a significant increase in turbulence further downstream. In addition to the TKE and the transport, would be nice to have the turbulence intensity TI values (as listed later on page 20, 10% for the baseline case), both for the baseline case and the maximum added TI reported - possibly somewhere around Figure 4. That again would give an indication on the applicability compared to the field values observed. Also note the typo in the caption of Figure 4: after c) all the subplots are marked to be continuously c).*

**Response:**

We have added the requested TI values in the discussion around Figure 4 as follows (p. 5, line 20):

" .., for which an enhanced recovery was found as discussed above. **The turbulence intensity** $TI \equiv (2k/3)^{1/2}/U_\infty$ **at hub height (not shown in the figure) is 10% at the inlet for both the reference case and the controlled case. The combination of reduced near-wake mean velocities and increased velocity fluctuations in the controlled case increase local** $TI$ **in the turbine wakes (ranging from** $\approx 2$**%-points in the wakes of middle rows to** $\approx 12$**%-points in the first and last rows). This increase in turbulence intensity dissipates to below** $1$**%-point difference at 10**$D$ **downstream of the last row.** "

Furthermore, we thank the reviewer for pointing out the typos in the caption. This has been fixed in the revised manuscript.

..................................................................................................

**4. Reviewer:** *On page 10, around line 10, the argument of "upstream actions do not require a specific downstream response in order to increase power in that downstream row", which is also paraphrased in the conclusions, needs to be elaborated. This rather broad conclusion seem to oversee the probability of the curtailment of the downstream turbine where down-regulation might be inevitable for certain CT set-points assigned to downstream turbine(s) in the resulting optimization. Could be partially true for the investigated C3t5 case since there observed very limited curtailment even at the most upstream turbine (as in Figure 3b). However, also seen in Figure 8b (except of the very last row as the authors indicated), there seem to be still a difference between on the power gain at turbine R11 for the scenarios of R1-R10 and R1-R11. Narrowing the argument to the considered case or very little to no downstream curtailment CT distributions is suggested.*

**Response:** We find it difficult to follow the argument formulated by the reviewer here. However, we agree in general that statements and conclusions made throughout section 3 are specific to the current case and should therefore be interpreted with care.

We have updated the manuscript to be more explicit in the sense that results are not

overall conclusions for wind-farm control in general, but should be interpreted as observations of the current optimal control case. See the beginning of Section 3 (p. 8, line 2):

" ... to uncover some of the characteristics of these control signals. **Note that the conclusions drawn within this section should be interpreted as observations of the current C3t5 optimal control cases, given specific wind-farm layout and flow conditions, and hence cannot just be generalized for any wind-farm control in general.** "

. . . . . . . . . . . . . . . . . . . . . . . . . . . . . . . . . . . . . . . . . . . . . . . . . . . . . . . . . . . . . . . . . . . . . . . . . . . . . . . . . . . . . .

**5. Reviewer:** *On page 13, line 13, "the presence of the flow invariant features of the control signals" needs further justification as Figure 11 would also depend on how variant the flow features are in the simulations. That should include both the spatial and temporal variance within the 30-min window. As far as the field measurements are concerned, high spatial and temporal correlations are observed. For the former, Figure 2 gives a brief idea about the wind speed range between the columns, that can be referred here. For the latter, time series or relevant temporal statistics can be presented to assess the randomness and strengthen the hypothesis.*

**Response:** We thank the reviewer for this useful comment. We have quantified spatial variance (between columns) as well as temporal variance (between time windows) of the flow field by investigating the correlation between the incoming disk-averaged velocity at $6D$ upstream of the first-row turbines (illustrated for two columns of the wind farm in Figure 1 of the current document). As shown in the Figure, there is considerable variation of the incoming velocities between columns and between time windows, and velocity fluctuations seem qualitatively uncorrelated. To keep the discussion in the manuscript concise, we do not include the figure below but instead report the average Pearson correlation coefficient between the incoming velocity fluctuations in different columns (for the full time horizon) and in different time windows (averaged over all

columns), with low values of 0.12 and 0.07 respectively.

We have incorporated this in the revised manuscript as follows (p. 14, line 4):

" ... the column swap is performed in 2 random independent ways. **The variability of flow conditions for different columns can be qualitatively observed in Figure 2. To further strengthen the hypothesis of the current experiment, we verified that the correlation between flow conditions in different columns is small, i.e. with an average Pearson correlation coefficient of 0.12 between columns for the incoming velocity fluctuations $6D$ upstream of the first row.** "

and (p. 14, line 11):

" whereas the time synchronization of control actions to specific flow events is eliminated. Similar to the first case, this is done in 2 random ways, **and the limited correlation between velocity fluctuations in different time windows was quantified at 0.07**. Figure 11b illustrates the row-averaged power for these... "

. . . . . . . . . . . . . . . . . . . . . . . . . . . . . . . . . . . . . . . . . . . . . . . . . . . . . . . . . . . . . . . . . . . . . . . . . . . . . . . . . . . .

**6. Reviewer:** *On page 17, around line 5, a very nice example on how to implement the optimized sinusoidal CT is presented. The practical examples can be further improved by a short discussion on the expected response time of such increases in tip speed ratio on a machine with high inertia. That would put the estimated sine wave period into perspective as well.*

**Response:** We thank the reviewer for this suggestion. Dynamic reductions from the maximum $C_T' = 3.5$ could, for instance, be achieved by using the fast pitch actuators with which modern turbines are equipped. We have added this comment to the discussion as follows (p. 17, line 25):

" **For instance, considering the NREL 5MW blade profiles, the maximum thrust coefficient of 3.5 can be attained by slightly changing the rotor design, e.g. using a 50% increase in blade chord length and an operational tip speed ratio 25%**

**higher than the original design value (see Goit and Meyers, 2015, Appendix A). Furthermore, given such redesign, dynamic reductions from this value could be realized through blade pitch control, for which actuation rates in the order of $10°/$s are possible (see, e.g., Jonkman et al. 2009). "**

Note that this control strategy is not unique and is solely given as an indication for the technical feasibility of tracking the proposed $C_T'$ waveform. In practice, we expect that this can also be achieved through a combination of generator torque and blade pitch control. This is subject of ongoing investigation.

. . . . . . . . . . . . . . . . . . . . . . . . . . . . . . . . . . . . . . . . . . . . . . . . . . . . . . . . . . . . . . . . . . . . . . . . . . . . . . . . . . . . . . . . . . . .

**7. Reviewer:** *For Section 4.2.3, the header "Full-scale wind farm test" is a bit misleading... Suggest to change to "Full-scale wind farm simulations (in LES)" instead.*

**Response:** Agreed, we have changed the section header to "Full-scale wind-farm LES"

. . . . . . . . . . . . . . . . . . . . . . . . . . . . . . . . . . . . . . . . . . . . . . . . . . . . . . . . . . . . . . . . . . . . . . . . . . . . . . . . . . . . . . . . . . . .

**8. Reviewer:** *On Figure 19, why would the power decrease after Row 5 for the sinusoidal case?*

**Response:** This can be explained based on Figure 20: it can be seen from the cross-section views that by sinusoidally varying first-row thrust, the time-averaged axial velocity at the turbine disk for R2 is increased, but the flow above the turbine disk is actually slowed down, hence indicating that momentum is entrained from the internal boundary layer above the turbine canopy. In downstream rows, this high momentum zone expands and mixes with the background flow field. Starting from R5, this zone is almost completely dissipated, and the disk velocity is actually slighter lower than in the reference case. In short, the sinusoidal control actions of the first row actually entrain momentum from the surrounding flow that would otherwise be entrained by natural turbulent mixing further downstream in the wind farm.

Moreover, it can be seen that the full optimal control (C3t5 in Figure 20) is able to contain the high momentum zone by continuously improving the mixing process with the background boundary layer. The mechanisms thereof however remain elusive to date, and are subject of further investigation.

We have incorporated this discussion in the revised version of the manuscript as follows (p. 23, line 8):

" ... with higher rotor velocities in the downstream as well. **Note also that, for the fifth row, the disk velocity is slightly lower for the sinusoidal control case than for the reference case, consistent with the decreased power extraction observed in Fig. 19. This can be explained by the fact that first-row control actions cause enhanced entrainment of momentum from the internal boundary layer above the turbine canopy that would otherwise be entrained by natural turbulent mixing in passive downstream rows. In consequence, lesser entrainment occurs for downstream rows, resulting in a slight decrease in disk velocities from the fifth row onwards.** "

. . . . . . . . . . . . . . . . . . . . . . . . . . . . . . . . . . . . . . . . . . . . . . . . . . . . . . . . . . . . . . . . . . . . . . . . . . . . . . . . . . . . .

**9. Reviewer:** *Page 22 around line 5, the (inevitable) discussions on loads are included. In addition to the loads on the controlled upstream turbine, Figure 20(b) indicates partial wakes on the further downstream rows of turbines. Therefore, the section should be improved by highlighting the probable increase in fatigue loading for not just turbine(s) R1 but for the downstream rows as well, possibly starting as early as R3.*

**Response:** We thank the reviewer for this comment. Agreed, the downstream turbines could also be subjected to increased fatigue loading, and we have included this in the revised manuscript as follows (p. 24, line 4):

" ... to fatigue loading **of the first-row turbines. Furthermore, partial wake alleviation and unsteady passing of abovementioned vortex rings could also increase**

**fatigue loading in downstream rows.** Hence, structural aspects ... "

..............................................................................................................

**10. Reviewer:** *On the grammatical note, the manuscript is clear and easy to follow.*
*The only comment might be on the use of Sect. or Section; Fig. or Figure references.*

**Response:** We have followed the official guidelines for referring to figures and sections as provided by the publisher, paraphrasing from the website:

*The abbreviation "Sect." should be used when it appears in running text and should be*
*followed by a number unless it comes at the beginning of a sentence.*
(https://www.wind-energy-science.net/for_authors/manuscript_preparation.html)

————————————————————

**Fig. 1.** Disk-averaged axial velocity upstream of the first-row turbines for column 1 and column 2 in Figure 2 of the manuscript. Vertical dotted lines demarcate the optimization window boundaries.